# Reservoir-computing based associative memory and itinerancy for complex dynamical attractors

Ling-Wei Kong [1,2], Gene A. Brewer[3] & Ying-Cheng Lai [2,4] ✉

Traditional neural network models of associative memories were used to store and retrieve static patterns. We develop reservoir-computing based memories for complex dynamical attractors, under two common recalling scenarios in neuropsychology: location-addressable with an index channel and content-addressable without such a channel. We demonstrate that, for location-addressable retrieval, a single reservoir computing machine can memorize a large number of periodic and chaotic attractors, each retrievable with a specific index value. We articulate control strategies to achieve successful switching among the attractors, unveil the mechanism behind failed switching, and uncover various scaling behaviors between the number of stored attractors and the reservoir network size. For content-addressable retrieval, we exploit multistability with cue signals, where the stored attractors coexist in the high-dimensional phase space of the reservoir network. As the length of the cue signal increases through a critical value, a high success rate can be achieved. The work provides foundational insights into developing long-term memories and itinerancy for complex dynamical patterns.

While the development of artificial neural memories had been earlier, the celebrated Hopfield model[1,2] was a significant milestone in the modeling of associative memory in neural networks. In this model, the neurons are characterized by discrete dynamical states (e.g., +1 or −1). With training to adjust the connection weights among the neurons, the network can store a number of patterns represented by a grid of such discrete values. When a new pattern is presented to the network, it responds by setting the neural states in such a way that the resulting pattern most closely resembles the new one - the process of memory retrieval. The neural network thus constitutes effectively an associative memory. The dynamics of the neurons in the original Hopfield model are simply binary, but more sophisticated, oscillatory neural dynamical models can be used for associative memory[3–8]. Recently, deep neural networks have been used to implement associative memory[9]. It is noteworthy that, in most previous works on artificial neural networks as associative memory, the patterns to be "memorized" were static.

Can artificial neural networks be exploited as memory device for complex dynamical patterns, such as chaotic attractors? Such an object is the result of the time evolution of a nonlinear dynamical system, which manifests itself as a dynamical trajectory in the phase space. One might be tempted to represent a chaotic attractor using a grid of cells as a static pattern. There are two fundamental difficulties rendering this approach inappropriate. First, a chaotic attractor typically possesses a fractal structure in phase space[10], so representing it using a static grid may require a prohibitively high resolution. Second, the probabilities for the trajectory to land in different cells can be singular, defying using a finite set of discrete values to represent the "frequencies of visit" to different cells. Here, for realizing associative memory for complex dynamical attractors, we propose a drastically different approach than the Hopfield type of neural networks. The idea is to train a neural machine as a dynamical system with the capability to generate a variety of distinct dynamical trajectories, each

[1]Department of Computational Biology, Cornell University, Ithaca, New York, USA. [2]School of Electrical, Computer and Energy Engineering, Arizona State University, Tempe, Arizona, USA. [3]Department of Psychology, Arizona State University, Tempe, Arizona, USA. [4]Department of Physics, Arizona State University, Tempe, Arizona, USA. ✉e-mail: Ying-Cheng.Lai@asu.edu

corresponding to a complex attractor to be memorized. The purpose of this paper is to demonstrate that this is indeed feasible and, moreover, to study what phenomena would emerge in such multifunctional dynamical systems.

For the associative memory to produce the correct dynamical trajectory corresponding to a desired attractor, an essential requirement for successful retrieval of the attractor is that the memory itself be a closed-loop dynamical system capable of generating continuous time evolution of the relevant state variables. Reservoir computing[11–13], a class of recurrent neural networks (RNNs) that has been extensively investigated for model-free prediction of chaotic systems in recent years[14–38], stands out as a suitable choice. Quite recently, an implementation of neural computation through reservoir computing was demonstrated[39]. Reservoir computing is a particularly suitable framework for our purposes. The training process is highly efficient, requiring only a linear regression to achieve satisfactory results. This efficiency proves invaluable as the number of target states increases. More importantly, this straightforward training method helps to avoid several significant issues. Notably, it circumvents the problem of catastrophic forgetting, which is particularly challenging when neural networks need to memorize a large number of distinct states[40]. Additionally, it addresses the vanishing/exploding gradient problem that can hinder the learning process in RNNs. Here, we shall demonstrate how reservoir computing can be exploited to store and retrieve complex dynamical attractors. Different attractors are trained to coexist within a single reservoir neural network and, when needed, can be recalled later by suitable cues. For simplicity, a reservoir neural network is sometimes also called a reservoir computer (RC).

In the science of memory, there are two types of memories: short-term and long-term[41–45]. Reservoir computing, because of its recurrent structure, has naturally incorporated memories in its dynamical evolution in that past inner states and inputs can affect the future state and outputs. The temporary information of previous inputs stored within the dynamical state evolution of the hidden layer represents a kind of short-term memory of the RNN. The focus of our study is on long-term memories that are encoded within the weights and connections in the neural network architecture, manifested as the stabilized dynamical trajectories that can be maintained in the dynamical network. In neuropsychology, two distinct types of models of long-term memory are often used: "location-addressable" and "content-addressable" memories[46], where the cue used to address the target memory for the former corresponds to a specific index for each memorized pattern and, for the latter, the cue can be made of a short time series correlated with the dynamical trajectory from the memorized attractor. Our machine-learning memory for complex dynamical attractors incorporates both types of models, named index-based and index-free memory, respectively.

A neural network capable of memorizing and retrieving multiple memory states is also a multifunctional neural network. When different states are recalled, the neural network exhibits distinct dynamical behaviors. The idea of multifunctional RNNs has been discussed previously with different forms of implementation. In particular, in the index-based approach, an index parameter is introduced to modulate the functionality of the neural networks and store different states with different index values[29,31,32,47,48], but recalling the memorized states was a challenge, for which control may be necessary. To our knowledge, prior to our work, there were no discussions about the scaling law of the memory capacity in the context of storing different states in RNNs. Exploiting multistability in the neural networks represents another approach - the index-free approach[49–52], where multiple attractors coexist in the high-dimensional hidden phase space of the reservoir network. In this regard, previous work focused on storing fixed points[49], and there were also methods based on storing chaotic or periodic states[50–52]. Outstanding issues included the dynamical mechanism underlying the retrieval of the coexisting dynamical states

and their basin structure. Here, we shall demonstrate that our reservoir-computing based classifier can distinguish among different recalled states and between a successful and failed recall, both with high accuracy, thereby providing quantitative insights into the open issues.

Our main results can be summarized, as follows. For the location-addressable scenario, a single reservoir computer can "memorize" a number of distinct complex dynamical attractors (also referred to as the complex memory states). Each memory state is embedded in the high-dimensional phase space of the hidden reservoir neural network, is maintained (stored) indefinitely, and can be retrieved whenever needed. A key issue is how successful a transition between two arbitrarily memorized states can be, for which some proper control actions are needed. We calculate the success rate of transition among different states in a reservoir computer, obtain a dynamical understanding behind the failed switching, and articulate several control strategies to significantly enhance the success rate. We also demonstrate the success of memorizing hundreds of different attractors in one single reservoir computer and uncover scaling laws between the size of the reservoir network and the number of memory states. For the content-addressable setting without an index channel, we exploit multistability for retrieval with the aid of some cue signals, where different memory states can be coexisting attractors or long transient states. We find that the stored states can indeed be recalled by short cues and even partial or noisy cues. We uncover a transition phenomenon that arises in the retrieval success rate as the length of the cue signal varies, as well as sophisticated basin structures in the hidden high-dimensional phase space of the reservoir network. A connection between the transition phenomenon of success rate and the basin structure is established, yielding a dynamical understanding of the mechanism behind memory retrieval from the coexisting states. We also find that a natural random itinerancy is possible when there is noise on the neurons in our reservoir computers. These results provide foundational insights into developing machine-learning-based long-term memory devices for complex dynamical states or attractors.

## Results

### Index-based reservoir-computing memory

In the "location-addressable" or "parameter-addressable" scenario, the stored memory states within the neural network are activated by a specific location address or an index parameter. The stimulus that triggers the system to switch states can be entirely unrelated to the content of the activated state, and the pair linking the memory states and stimuli can be arbitrarily defined. For instance, the stimulus can be an environmental condition such as the temperature, while the corresponding neural network state could represent specific behavioral patterns of an animal. An itinerancy among different states is thus possible, given a fluctuating environmental factor. (The "parameter-addressable" scenario is different from the "content-addressable" scenario that requires some correlated stimulus to activate a memory state.)

**Storage of complex dynamical attractors.** The architecture of our index-based reservoir computing memory consists of a standard reservoir computer (i.e., echo state network)[11–13] but with one feature specifically designed for location-addressable memory: an index channel, as shown in Fig. 1(A). During training, the time series from a number of dynamical attractors to be memorized are presented as the input signal $\mathbf{u}(t)$ to the reservoir machine, each is associated with a specific value of the index $p$ so that the attractors can be recalled after training using the same index value. This index value $p$ modulates the dynamics of the RC network through an index input matrix $W_{\text{index}}$ that projects $p$ to the entire hidden layer with weights defined by the matrix entries. The training process is an open-loop operation with input $\mathbf{u}(t)$ and the corresponding index value $p$ through the index channel. To store or memorize a different attractor, its dynamical trajectory

becomes the input, together with the index value for this attractor. After training, the reservoir output is connected with its input, generating closed-loop operation so that the reservoir machine is now capable of executing self-dynamical evolution starting from a random initial condition, "controlled" by the specific index value. More specifically, to retrieve a desired attractor, we input its index value through the index channel, and the reservoir machine will generate a dynamical trajectory faithfully representing the attractor, as schematically illustrated in Fig. 1(A) with two examples: one periodic and another

chaotic attractor. The mathematical details of the training and retrieval processes are given in Supplementary Information (SI).

To demonstrate the workings of our reservoir computing memory, we generate a number of distinct attractors from three-dimensional nonlinear dynamical systems. We first conduct an experiment of memorizing and retrieving six attractors: two periodic and four chaotic attractors, as represented by the blue trajectories in the top row of Fig. 1(B) (the ground truth). The index values for these six attractors are simply chosen to be $p_i = i$ for $i = 1, ..., 6$. During

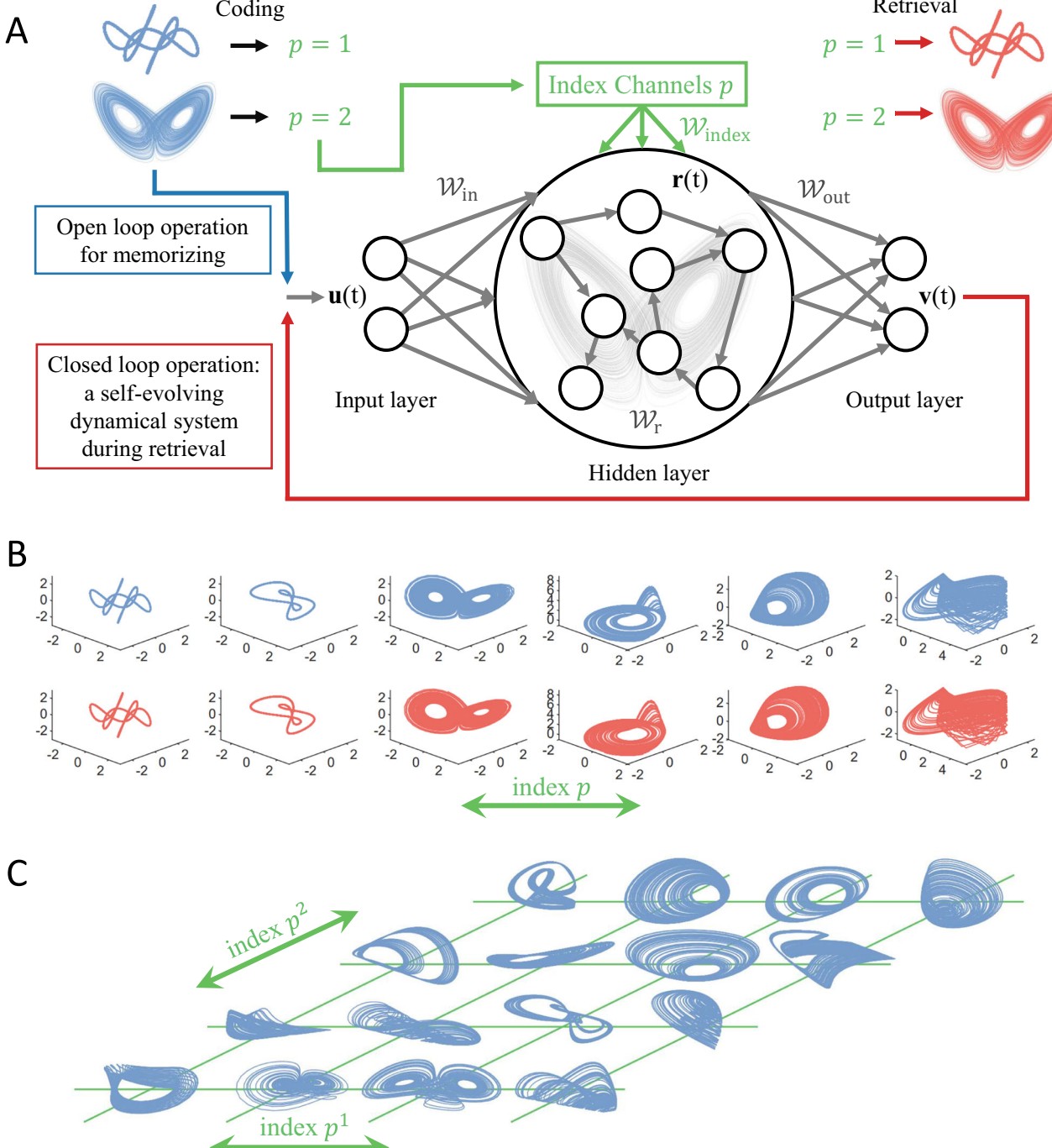

**Fig. 1 | Memorizing attractors with index-based reservoir computing. A** A schematic illustration of index-based reservoir-computing memory. **B** Retrieval of six memorized chaotic and periodic attractors using a scalar index $p_i$ (one integer value for each memorized dynamical attractor), where the blue and red trajectories in the top and bottom rows, respectively, represent the original and retrieved

attractors. Visually, there is little difference between the original and memorized/retrieved attractors. **C** Memorizing 16 chaotic attractors with two indices. The successfully retrieved attractors are presented in SI. All the memorized attractors are dynamically stable states in the closed-loop reservoir dynamical system.

training, the time series of each attractor is injected into the input layer, and the reservoir network receives the specific index value, labeling this attractor to be memorized so that the reservoir-computing memory learns the association of the index value with the attractor to be memorized. To retrieve a specific attractor, we set the index value $p$ to the one that labeled this attractor during training and allow the reservoir machine to execute a closed-loop operation to generate the desired attractor. For the six distinct dynamical attractors stored, the respective retrieved attractors are shown in the second row in Fig. 1(B). The recalled attractors closely resemble the original attractors and for any specific value of the index, the reservoir-computing memory is capable of generating the dynamical trajectory for an arbitrarily long period of time. The fidelity of the recalled states, in the long run, can be assessed through the maximum Lyapunov exponent and the correlation dimension. In particular, we train an ensemble of 100 different memory RCs, recall each of the memory states, and generate outputs continuously for 200,000 steps. The two measures are calculated during this generation process and compared with the ground truth. As an example, for the Lorenz attractor with the true correlation dimension of $2.05 \pm 0.01$ and largest Lyapunov exponent of about 0.906, 96% and 91% of the retrieved attractors have their correlation dimension values within $2.05 \pm 0.02$ and exponent values in $0.906 \pm 0.015$, indicating that the memory RC has learned the long-term "climate" of the attractors and is able to reproduce them (see SI for more results).

For a small number of dynamical patterns (attractors), a scalar index channel suffices for accurate retrieval, as illustrated in Fig. 1(B). When the number of attractors to be memorized becomes large, it is useful to increase the dimension of the index channel. To illustrate this, we generate 16 distinct chaotic attractors from three-dimensional dynamical systems whose velocity fields consist of different combinations of power-series terms[53], as shown in Fig. 1(C). For each attractor, we use a two-dimensional index vector $p_i = (p_i^1, p_i^2)$ to label it, where $p_i^1, p_i^2 \in \{1,2,3,4\}$. The attractors can be successfully memorized and faithfully retrieved by our index-based reservoir computer, with detailed results presented in SI.

Given a number of dynamical patterns to be memorized, there can be many different ways of assigning the indices. In addition to using one and two indices to distinguish the patterns, as shown in Fig. 1(B) and (C), we studied two alternative index-assignment schemes: binary and one-hot code. For the binary assignment scheme, we store $K$ dynamical patterns using $\log_2 K$ number of channels. The index value in each channel can either be one of the two possible values. For the one-hot assignment scheme, the number of index channels is the same as the number $K$ of attractors to be memorized. The index value in each channel is still one of the two possible values, for instance, either 0 or 1. For the one-hot assignment, each attractor is associated with one channel, where only this channel can take the number one for the attractor, while the values associated with all other channels are zero. Similarly, for each channel, the index value can be one if and only if the attractor being trained/recalled is the state with which it is associated. Arranging all the index values as a matrix, one-hot assignment leads to an identity matrix. For a given reservoir hidden-layer network, different ways of assigning indices result in different index-input schemes to the reservoir machine and can thus affect the memory performance and capacities. The effects are negligible if the number of dynamical attractors to be memorized is small, e.g., a dozen or fewer. However, the effects can be pronounced if the number $K$ of patterns becomes larger. One way to determine the optimal assignment rule is to examine, under a given rule, how the memory capacity depends on or scales with the size of the reservoir neural network in the hidden layer, which we will discuss later.

**Transition matrices among stored attractors.** We study the dynamics of memory toggling among the stored attractors. As each memorized state $s_i$ is trained to be associated with a unique index value $p_i$, the reservoir network's dynamical state among different attractors can be switched by simply altering the input index value. Several examples are shown in Fig. 2(A) and (B), where $p$ is switched among various values multiple times. The dynamical state activated in the reservoir network switches among the learned attractors accordingly. For instance, at step 800, for a switch from $p = 1$ to $p = 6$, the reservoir state switches from a periodic Lissajous state to a chaotic Hindmarsh-Rose (HR) neuron state. However, not all such transitions can be successful - a failed example is illustrated in the bottom panel of Fig. 2(C). In this case, after the index switching, the reservoir system falls into an undesired state that does not belong to any of the trained states. The probability of failed switchings is small and asymmetric between the two states before and after the switching. A weighted directed network can be defined among all the memorized states where the weights are the success probabilities, which can be represented by a transition matrix. Making index switches at many random time steps and counting the successful switches versus the failed switches, we numerically obtain the transition matrix, as exemplified in Fig. 2(D), from two different randomly generated indexed RCs. Figure 2(E) shows the average transition matrix of an ensemble of 25 indexed RCs. Figure 2(F) shows that the variance of the success rate across different columns (i.e., different destination states) is much larger than the variance across different rows (i.e., different starting states), implying that the success rate is significantly more dependent on the destination attractor than on the starting attractor.

What is the origin of the asymmetric dependence in the transition matrices and the dynamical mechanism behind the switch failures? Two observations are helpful. The first concerns the dynamic consequence of a switch in the index value $p$. Incorporating this term of $p$ into the reservoir computer is equivalent to adding an adjustable bias term to each neuron in the reservoir hidden layer. Different values of $p$ thus directly result in different bias values on the neurons and different dynamics. The same indexed RC under different $p$ can be treated as different dynamical systems. Second, note that, during a switch, one does not directly interfere with the state input $u$ or the hidden state $r$ of the RC network, but only changes the value of $p$. Consequently, a switch in $p$ as in Fig. 2(A) switches the dynamical equations of the RC network while keeping the RC states $r_{\text{last}}$ and the output $v_{\text{last}}$ at the last time step, which is passed from the previous dynamical system to the new system after switching. This pair of $r_{\text{last}}$ and $v_{\text{last}}$ becomes the initial hidden state and initial input under the new dynamic equation of RC.

The two observations suggest examining the basin of attraction of the trained state in the memory RC hidden space under the corresponding index value, which typically does not fully occupy the entire phase space. Figure 2(I) shows the basin structure of two arbitrary states in an indexed RC trained with 16 attractors. The blue regions, leading to the trained states, leave some space for the orange regions that lead to untrained states and failed to switch. If the RC states at the last time step before switching lay outside the blue region, the RC network will evolve to an unwanted state, and a switch failure will occur. We further plot the points from the attractor before switching in blue (successful) and orange (unsuccessful), as shown in Fig. 2(H). They are the projections of the basin structure of the new attractors onto the previous attractor. This picture provides an explanation for the strong dependence of success rates on the destination states. In particular, the success rate is determined by two factors: the relative size of the basin of the new state after switching and the degree of overlap between the previous attractor and the new basin. While the degree of overlap depends on both the starting and destination states, the relative size of the new state basin is solely determined by the destination state, resulting a strong dependency on the switching success rate of the destination state. [Further illustrations of (i) the basin structure of the memorized states in an indexed RC, (ii) projections of the basin structure of the destination state back to the starting

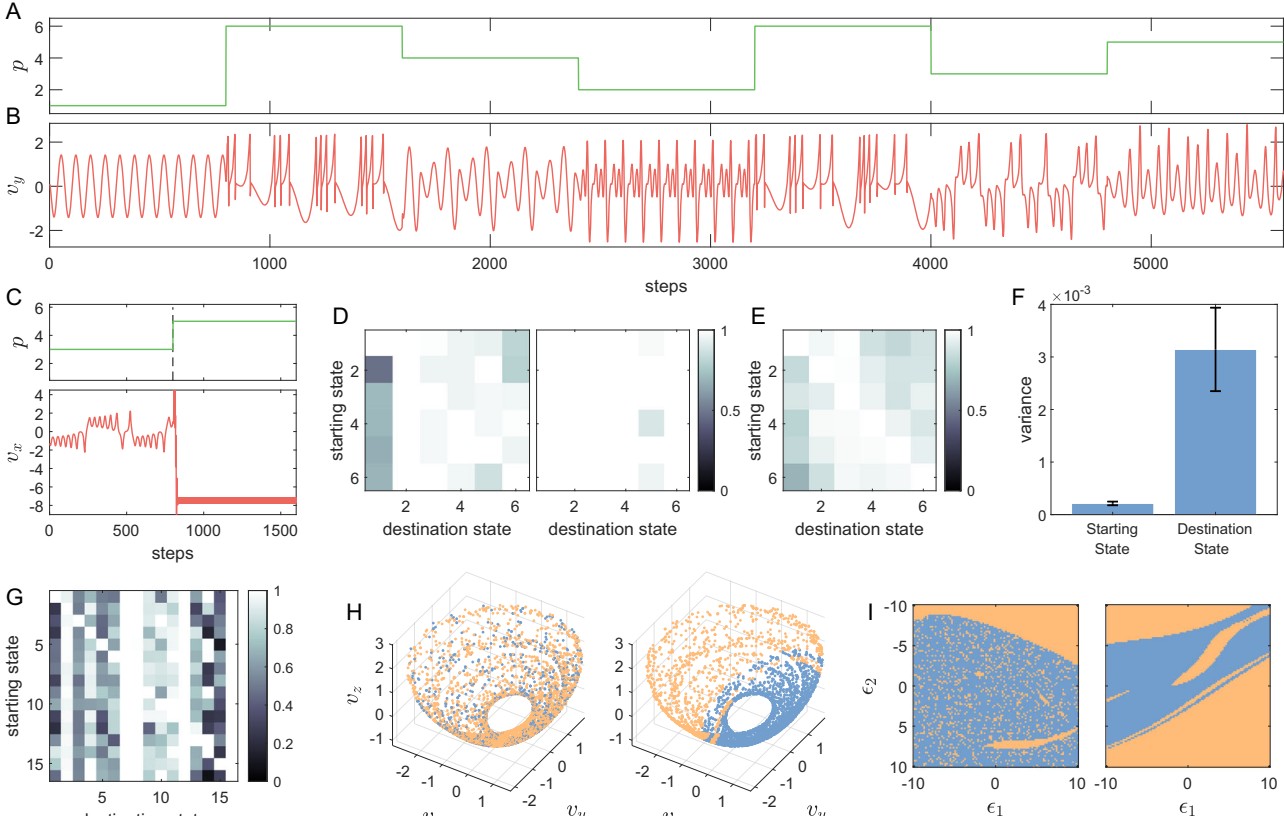

**Fig. 2 | Transitions among different memorized attractors in an indexed memory RC. A**, **B** By changing the index value $p$ as in panel (**A**), one can toggle among different states shown in panel (**B**) that have been memorized as attractors in the indexed memory RC (Dimensions $v_x$ and $v_z$ are not shown). **C** An instance of a failed switch, where the memory RC evolves to an untrained state after changing $p$. **D** Two typical transition matrices where six different states [as the ones in Fig. 1(B)] were memorized. **E** The average of transition matrices from 25 memory RCs with different random seeds and the same settings as in panel (**D**). **F** Variances in the success rate with respect to the starting and destination attractors, revealing a stronger dependence on the latter. The result is from the ensemble of 25 RC networks, each trained with ten different chaotic attractors. **G** A deliberately chosen memory RC with 16 memory states has relatively low success rates among many states for visualizing the basin structures shown below. **H** Attracting regions of two

different destination states on the manifold of a starting state. The memory RC originally operates on this shell-shaped starting attractor, and the index $p$ is toggled at different time steps when the memory RC is at different positions on this attractor. Such a position can determine whether the switching is successful, where the successful positions are marked blue and failed positions are marked lighter orange. **I** Local two-dimensional slices of the basin structures of the two destination states from panel (**H**) in the high-dimensional RC hidden space. Again, the blue and lighter orange regions mark the basins of the memorized state and of some untrained states leading to failed switches, respectively. The quantities $\epsilon_1$ and $\epsilon_2$ define the scales of perturbations to the RC hidden space in two randomly chosen perpendicular directions. The $\epsilon_1 = \epsilon_2 = 0$ origin points are the RC hidden states taken from random time steps while the memory RC operates at the corresponding destination states.

state attractor, and (iii) how switch success is strongly affected by the basin structure of the destination state are provided in SI.]

**Control strategies for achieving high memory transition success rates.** For a memory system to be reliable, accessible, and practically useful, a high success rate of switching from one memorized attractor to another upon recalling is essential. As this switching failure issue is rooted in the problem of whether an initial condition is inside the attracting basin of the destination state, such type of failure can be universal for models for memorizing not exactly the (static) training data but the dynamical rules of the target states and re-generating the target states by evolving the memorized dynamical rules. Even if one uses a series of networks to memorize the target states separately, the problem of properly initializing each memory network during a switch or a recall persists. A possible solution to this initial state problem is to optimize the training scheme or the RNN architecture to make the memorized state globally attractive in the entire hidden space - an unsolved problem at present. Here we focus on noninvasive control strategies to enhance the recall or transition success rates without altering the training technique or the RC architecture, assuming a memory RC is already trained and fixed. The

established connections or weights will not be modified, ensuring that the memory attractors already in place are preserved.

The first strategy, named "tactical detour," utilizes some successful pathways in the transition matrix to build indirect detours to enhance the overall transition success rates. Instead of switching from an initial attractor to a destination attractor directly, going through a few other intermediate states can result in a high success rate, as exemplified in Fig. 3(A). This method can be relatively simple to implement in many scenarios, as all needed is switching the $p$ value a few more times. However, this strategy has two limitations. First, it is necessary to know some information about the transition matrix to search for an appropriate detour and estimate the success rate. Second, the dynamical mechanism behind failed switching suggests that the success rate mostly depends on the relative size of the attracting basin of the destination state, implying that a state that has a small attracting basin is difficult to reach from most other states. As a result, the improvement of the success rate from a detour can fall below 100%. As exemplified in Fig. 3(B), the success rate with detours, which can significantly increase the overall success rate within just a few steps, can saturate at some levels lower than one.

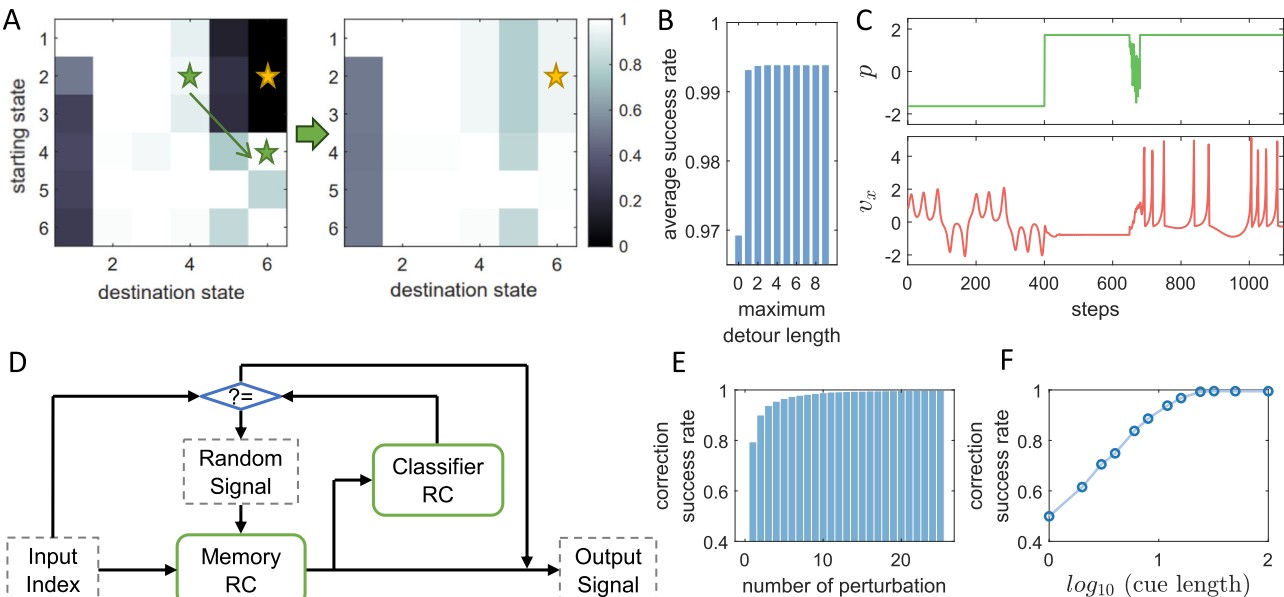

**Fig. 3 | Control strategies for achieving high memory transition success rates.**
**A** Achieving higher switching success rates using detours, where the switching from attractor 2 to attractor 6 via a detour through attractor 4 significantly increases the success rate. The right panel illustrates the improved transition matrix with detours as compared to the original one on the left. **B** Average success rate with different maximum detour lengths. A minimal detour is typically needed - just one single step of detour can reduce more than two-thirds of the failing probability, obtained from the same ensemble of memory RCs as in Fig. 2(F). **C** An instance of feedback control to correct a failed switch, following the scheme in panel (**D**). The switch at the 400th time step leads to an untrained static state. Since the classifier RC cannot recognize the memory RC output as the desired state, a short random signal is applied to perturb the memory RC, which adjusts the state of the memory RC to achieve a successful retrieval. **D** Illustration of the controlled memory retrieval by a classifier RC and random signals, where a classifier RC is used to identify the output from the memory RC. The outcome of this identification decides whether to apply a random perturbation or to use the memory RC output as the final result. **E** The rates of successfully corrected failed switches by the control mechanism illustrated in (**C**) and (**D**). The denominator of this correction success rate is the number of failed switches, not all switches. Each perturbation is implemented by ten steps of standard Gaussian noise. Just one "perturb and classify" cycle can correct about 80% of the failed switches. The results of this approach with different parameter choices are shown in SI. **F** Performance of the cue-based method to correct failed switches with respect to cue length. Within 25 steps of the cue, the correction success rate reaches 99%. Panels (**E**) and (**F**) are generated from the same ensemble of 25 memory RCs, each trained to memorize the six attractors as in Fig. 1.

Our second strategy is of the feedback control type with a classifier reservoir computer and random signals to perturb the memory reservoir computer until the latter reaches the desired memory state, as illustrated and explained in Fig. 3(C, D). This trial-and-error approach is conceptually similar to the random memory search model in psychology[54]. A working example is shown in Fig. 3(C), where the input index $p$ first activates certain outputs from the memory RC, which are fed to a classifier reservoir computer trained to distinguish among the target states and between the target states and non-target states. As detailed in SI, a simple training of this classifier RC can result in a remarkably high accuracy. The output of the classifier RC is compared with the index value. If they agree, the output signal represents the correct attractor; otherwise, a random signal is injected into the memory RC to activate a different outcome. The feedback loop continues to function until the correct attractor is reached. This control strategy can lead to a nearly perfect switch success rate, at the cost of a possibly long switching time. Let $T_n$ be the time duration that the random perturbation lasts, $T_c$ be the time for making a decision about the output of the classifier, and $\eta_{i,j}$ be the switching success rate from a starting attractor $s_i$ to a destination attractor $s_j$. The estimated time for such a feedback control system to reach the desired state is $T_{\text{reach},i,j} = (T_n + T_c)/\eta_{i,j}$, which defines an asymmetric distance between each pair of memorized attractors.

To provide a quantitative estimate of the performance of this strategy, we run 90,000 switching in 25 different memory RCs trained with the six periodic and chaotic states as illustrated in Fig. 1(B). The size of the reservoir network is 1000, and other hyperparameter values are listed in SI. Among the 90,000 switchings, 8110 trials (about 9%)

failed, which we used as a pool to test the control strategies. To make the results generic, we test 12 different control settings with different parameters. We test four different lengths of each noisy perturbation period, including 1 step, 3 steps, 10 steps, and 30 steps, with Gaussian white noise at three different levels (standard deviations: $\sigma = 0.3$, $\sigma = 1$, and $\sigma = 3$). We find that with an appropriate choice of the control parameters, 10 periods of 10 steps of noise perturbations (so 100 steps of perturbations in total) can eliminate 99% of the failed switchings, as illustrated in Fig. 3(E). The full results of the 12 different control settings are demonstrated and discussed in SI.

Our third control strategy is also motivated by daily experience: recalling an object or an event can be facilitated by some relevant, reminding cue signals. We articulate using a cue signal to "warm up" the memory reservoir computer after switching the index $p$ to that associated with the desired attractor to be recalled. The advantage is that the memory access transition matrix is no longer needed, but the success depends on whether the cue is relevant and strong enough. The cue signals are thus state-dependent: different attractors require different cues, so an additional memory device may be needed to restore the warming signals containing less information than the attractors to be recalled. This leads to a hierarchical structure of memory retrieval: (a) a scalar or vector index $p$, (b) a short warming signal, and (c) the whole memorized attractor, similar to the workings of human memory suggested in ref. 55. We also examine if state-independent cues (uniform cues for all the memory states) can be helpful, and find that in many cases, a simple globally uniform cue can make most memorized attractors almost randomly accessible, but there is also a probability that this cue can almost entirely block a few memory states.

be eliminated with just 8 steps of cue, and 24 steps of cue can eliminate more than 99% of the failed switchings.

**Scaling law for memory capacity.** Intuitively, a larger RNN in the hidden layer would enable more dynamical attractors to be memorized. Quantitatively, such a relation can be characterized by a scaling law between the number $K$ of attractors and the network size $N$. Our numerical method to uncover the scaling law is as follows. For a fixed pair of $(K, N)$ values, we train an ensemble of reservoir networks, each of size $N$, using the $K$ dynamical attractors. The fraction of the networks with validation errors less than an empirical threshold gives the success rate of memorizing the patterns, which in general increases with $N$. A critical network size $N_c$ can be defined when the success rate is about 50%. (Here, we use 50% as a threshold to minimize the potential error in $N_c$ due to random fluctuations. In SI, we provide results with 80% as the success rate threshold. The conclusions remain the same under this change.)

We use three different types of validation performance measures. The first is the prediction horizon defined by the prediction length before the deviation of the prediction time series reaches 10% of the oscillation amplitude of the real state (half of the distance of the maximum value minus the minimum value in a sufficiently long time series of that target state). The oscillation amplitude and the corresponding deviation rate are calculated for each dimension of the target state, and the shortest prediction horizon is taken among all the dimensions of the target state. The prediction horizon is rescaled by the length of the period of the target system to facilitate comparison. For a chaotic state, we use the average period defined as the average shortest time between two local maximums in a sufficiently long time series of that target state. The threshold of this prediction horizon for a successful recall is set to be two periods of each memory state. The second performance measure is the mean square root error (RMSE) with a validation time of four periods (or average periods). The third measure is defined by the time before the predicted trajectories go beyond a rectangular-shaped region surrounding the real memory state to an undesired region in the phase space, which is set to be 10% larger than the rectangular region defined by the maximum and minimum values of the memory state in each dimension. For these three measures, the thresholds for a successful recall can be different across datasets but are always fixed within the same task (the same scaling curve). The thresholds for the prediction-horizon-based measures and region-based measures are several periods of oscillation (or average periods for chaotic states) of the target state.

Figure 4(A) shows the resulting scaling law for the various tasks, with different datasets of states, different training approaches, and different coding schemes for the indexed RCs. The first dataset (dataset #1) consists of many thousands of different periodic trajectories. With index-based RCs, we use one-hot coding, binary coding, and two-dimensional coding with this dataset, as shown in Fig. 4(A). They all lead to similar algebraic scaling laws $N_c \propto K^\gamma$ that are close to a linear law: $\gamma = 1.08 \pm 0.01$ for both the one-hot coding and binary coding, and $\gamma = 1.17 \pm 0.02$ for the 2D coding. In all the three cases, $N_c$ all grow slightly faster than a linear law. A zoom-in picture comparing the three coding schemes is shown in Fig. 4(B). It can be seen that the one-hot coding is more efficient than the other two. Figure 4(B) also demonstrates how the data points in the scaling laws in Fig. 4(A) are gathered. Similar success rates versus $N$ curves are plotted for each $K$ value for each task and the data point from that curve which is closest to a 50% success rate is taken to be $N_c$.

Why is the one-hot coding more efficient than the other coding scheme we test? In our index-based approach to memorizing dynamical attractors, an index value $p_i$ is assigned for each attractor $s_i$ and acts as a constant input through the index channel to the reservoir neural network, which effectively modulates the biases to the network. Since the index input $p_i$ is multiplied by a random matrix before

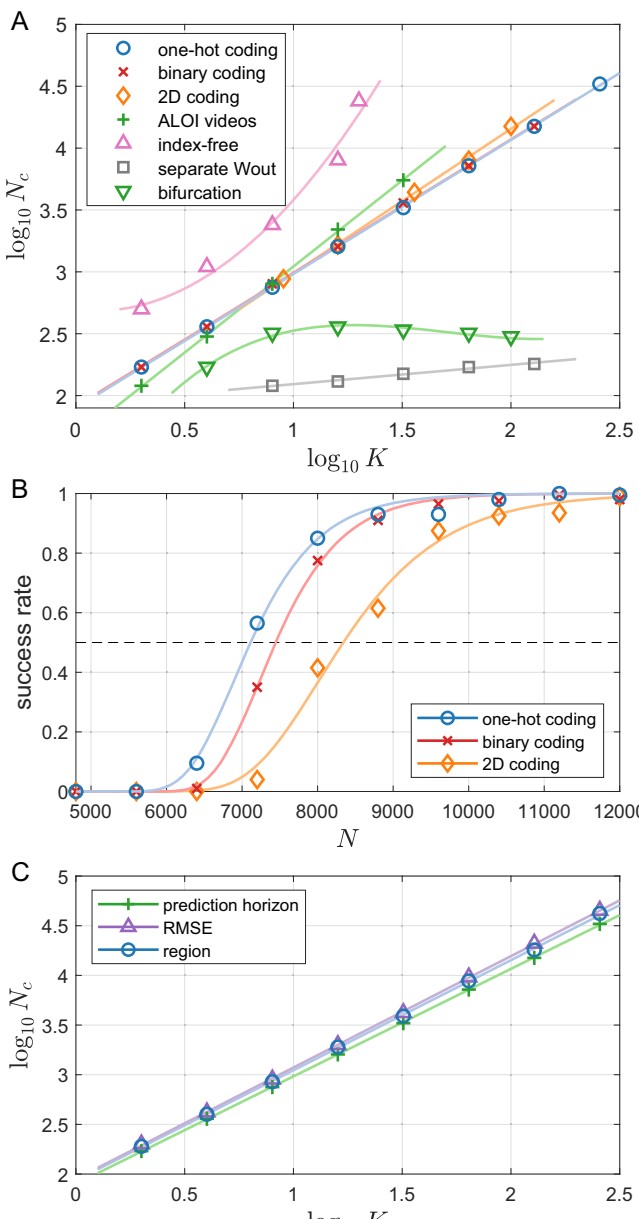

**Fig. 4 | Scaling behaviors of the memory capacity. A** Algebraic scaling relation between the number $K$ of dynamical patterns to be memorized and the required size $N_c$ of the RC network for various tasks and different memory coding schemes. Details of each curve/task are discussed in the main text. The "separate $W_{\text{out}}$" and "bifurcation" tasks are shown with prediction-horizon-based measures, while all the other curves are plotted with the region-based measure. **B** Examples of how the success rate increases with respect to $N$ and comparisons among different coding schemes, where the success rate of accurate memory recalling (using the region-based performance measure) of three coding schemes on the dataset #1 with $K = 64$ is shown. All three curves have a sigmoid-like shape between zero and one, where the data points closest to the 50% success rate threshold correspond to $N_c$. All data points in the scaling plots are generated from this type of curve (success rate versus $N$) to extract $N_c$. The one-hot coding is the most efficient coding of the three we tested for this task. **C** Comparisons among different recalling performance measures. The three different measures on the dataset #1 with a one-hot coding have indistinguishable scaling behavior with only small differences in a constant factor.

With this third control strategy with the cues, we again employ the 8,110 failed switching pool to test the strategy performance. As shown in Fig. 3(F), with the increase of the cue length, the success rate increases nearly exponentially. Almost 90% of the failed switchings can

entering the network, each neuron receives a different random level of bias modulation, affecting its dynamical behavior through the nonlinear activation function. In particular, the hyperbolic tangent activation function has a number of distinct regions, including an approximately linear region when the input is small, and there are two nearly constant saturated regions for large inputs. With one specific $p_i$ value injected into the index channel, different neurons in the hidden layer can be distributed to/across different functional regions (detailed demonstration in SI). When the index value is switched to another one, the functional regions for each neuron are redistributed, leading to a different dynamical behavior of the reservoir computer. Among the three coding schemes, the one-hot coding scheme utilizes one column of the random index input matrix, so the bias distributions for different attractors are not related to each other, yielding a minimal correlation among the distributions of the functional regions of the neurons for different values of $p_i$. For the other two coding schemes, there is a certain degree of overlap among the input matrix entries for different attractors, leading to additional correlations in the bias distributions, thereby reducing the memory capacity.

To make the stored states more realistic, we apply a dataset processed from the ALOI videos[56]. This dataset contains short videos of rotating small everyday objects, such as a toy duck or a pineapple. In each video, a single object rotates horizontally through a full 360-degree turn, returning to its initial angle. Repeating the video generates a periodic dynamical system. To reduce the computational loads due to the high spatial dimensionality of the video frames, we perform dimension reduction to the original videos through the principal component analysis, noting that most pixels are black backgrounds and many pixels of the object are highly correlated. We take the first two principal components of each video as the target states. The results are shown in Fig. 4(A) as the "ALOI videos" task. Similar to the previous three tasks, we obtain an algebraic scaling law $N_c \propto K^\gamma$ that is slightly faster than a linear law with $\gamma = 1.39 \pm 0.01$.

The "bifurcation" task is an example to show the potential of our index-based approach, where $N_c$ grows much slower than a linear scale with respect to $K$ and can even decrease in certain cases. It consists of 100 dynamical states gathered from the same chaotic food chain system but with different parameter values. We sweep an interval in the bifurcation diagram of the system with both periodic states of different periods and chaotic states with different average periods. For index coding, we use a one-dimensional index and assign the states by sorting the system parameter values used to generate these states. The index values are in the interval $[-2.5, 2.5]$, and are evenly separated on this interval. As shown in Fig. 4(A), the scaling behavior of this "bifurcation" task is much slower than a linear law, suggesting that when the target states are correlated, and the index values are assigned in a "meaningful" way, the reservoir computer can utilize the similarities among target states for more efficient learning and storing. Moreover, we notice that $N_c$ can even decrease in a certain interval of $K$, as the amount of total training data increases with larger $K$, making it possible for the reservoir computer to reach a similar performance with a smaller value of $N_c$. In other tasks with other training sets, training data from different target states are independent of each other. In summary, training with the index-based approach on this dataset is more resource-efficient than having a series of separate reservoir computers trained with each target state and adding another selection mechanism to switch the reservoir computer while operating switching or itinerary behaviors.

One of the most important features of reservoir computing is that the input layer and the recurrent hidden layer are generated randomly and, thus, are essentially independent of the target state. One way to utilize this convenient feature is to use the same input and hidden layer for all the target states but with a separate output layer trained for each target state. In such an approach, an additional mechanism of selecting the correct output layer (i.e., $W_{out}$) during retrieval is necessary, unlike

the other approaches studied with the same integrated reservoir computer for all the different target states. However, training with separate $W_{out}$ can lower the computational complexity in some scenarios, as shown by the "separate $W_{out}$" task in Fig. 4(A), where $N_c$ increases as a power law with $K$ that is much slower than a linear law. However, training with separate $W_{out}$ still makes the overall model complexity grow slower than linear, as the same number of independent $W_{out}$ is needed as the number of $K$. For a scenario such as in the "bifurcation" task, the vanilla index-based approach is more efficient. As the input layer and hidden layer are shared by all the different target states, the issue of failed switching still exists, so control is required.

We further demonstrate the efficiency of our index-based approach compared to the index-free approach. In the "index-free" task shown in Fig. 4(A), dataset #1 was used. While the index-based approach has an almost linear scaling, the scaling of the index-free approach grows much faster than a power law. For instance, the critical network size $N_c$ with $K = 20$ is about $N_c = 2.4 \times 10^4$, which is almost as large as the $N_c$ for $K = 256$ with the index-based memory RC on the same dataset. Moreover, the critical network size $N_c$ for $K = 32$ is larger than $10^5$. The comparison of the two approaches on the same task reveals that, while the index-free memory RC has the advantage of having content-addressable memory, it is computationally costly compared with the index-based approach, due to the severe overlapping among different target states for large $K$ values, thereby requiring a large memory RC to distinguish different states.

To test the genericity of the scaling law, we compare the results from the same task (the "one-hot coding" task) with the three different measures, as shown in Fig. 4(C). All three curves have approximately the same scaling behavior, except a minor difference in a constant factor.

## Index-free reservoir-computing based memory - advantage of multistability

The RC neural network is a high-dimensional system with rich dynamical behaviors[49–52,57,58], making it possible to exploit multistability for index-free memory. In general, in the high-dimensional phase space, various coexisting basins of attraction can be associated with different attractors or dynamical patterns to be memorized. As we will show, this coexisting pattern can be achieved with a similar training (storing) process to that of index-based memory but without an index. The stored attractors can then be recalled using the content-addressable approach with an appropriate cue related to the target attractor. To retrieve a stored attractor, one can drive the networked dynamical system into the attracting basin of the desired memorized attractor using the cues, mimicking a spontaneous dynamical response characteristic of generalized synchronization between the driving cue and the responding RC[50,59,60].

While the possibility of index-free memory RC was pointed out previously[50–52], a quantitative analysis of the mechanism of the retrieval was lacking. Such an analysis should include an investigation into how the cue signals affect the success rate, what the basin structure of the reservoir computer is, and a dynamical understanding connecting these two. A difficulty is how to efficiently and accurately determine, from a short segment of RC output time series, which memory state is recalled and if any target memory has been recalled at all. For this purpose, short-term performance measures such as the RMSE and prediction horizon are not appropriate, as the ground truth of a specific trajectory of the corresponding attractor is not available, especially for the chaotic states. Moreover, a measure based on calculating the maximum Lyapunov exponents or the correlation dimensions may not be suitable either, due to the requirement of long time series. In realistic scenarios, persisting the memory accurately for dozens of periods can be sufficient, without the requirement for long-term accuracy.

Our solution is to train another RC network (classifier RC) to perform the short-term classification task to determine which

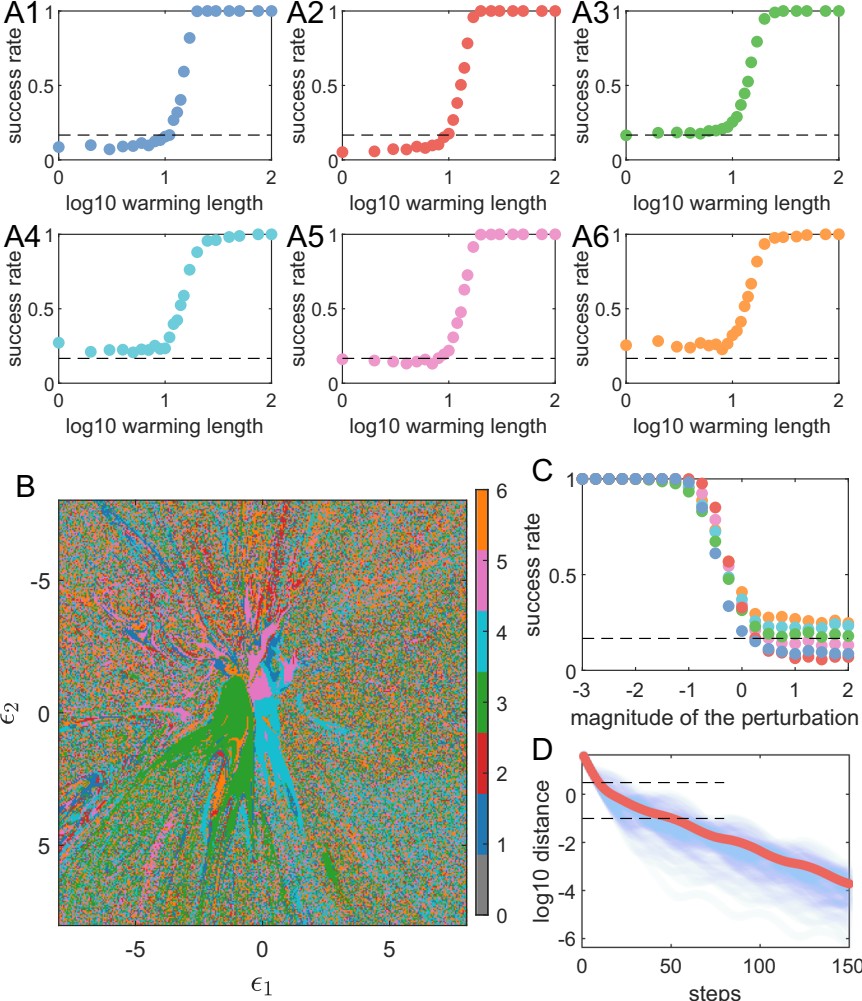

**Fig. 5 | Index-free memory recalling. A1–A6** Success rate of memory retrieval of the six attractors in Fig. 1(**B**) versus the length of the warming cue. In the 3D phase space of the original dynamical systems, these attractors are located in approximately the same region and are overlapping. **B** A 2D projection of a typical basin structure of the reservoir dynamical network, where different colors represent the initial conditions leading to different memorized attractors. The central regions have relatively large contiguous areas of uniform color, while the basin structure is fragmented in the surrounding regions. **C** Success rate for the memory reservoir network to keep the desired attractor versus the magnitude of random perturbation. The two plateau regions with ~100% and 1/6 success rates correspond to the two typical basin features shown in panel (**B**). **D** Distance (averaged over an ensemble) between the dynamical state of the reservoir neural network and that of the target attractor versus the cue duration during warming. The cyan curves are obtained from 100 random trials, with the red curve as the average. The two horizontal black dashed lines correspond to the perturbation magnitude of $10^{0.5}$ and $-1$ in (**C**), and their intersection points with the red curve represent the warming lengths of 10 and 50-time steps, respectively, which are consistent with the transition regions in (**A**), suggesting a connection between the basin structure and the retrieval behavior.

memorized states have been recalled or, if no successful recall has been made at all, with high classification accuracy. We have tested its performance on more than a thousand trials and found only 6 trials out of 1,200 trials of deviation from a human labeler (details in SI), which are intrinsically ambiguous to classify. The classifier RC is also used in the control scheme for index-based memory RC, as shown in Fig. 3(D). The classifier RC provides a tool to understand the retrieval process, e.g., the type of driving signals that can be used to recall a desired memorized attractor. The basic idea is that the cues serve as a kind of reminder of the attractor to be recalled, so a short period of the time series from the attractor serves the purpose. Other cue signals are also possible, such as those based on partial information of the desired attractor to recall the whole or using noisy signals for warming.

We use the six different attractors in Fig. 1(B) as the target states, all located in a three-dimensional phase space. They do not live in distinct regions but have significant overlaps. However, the corresponding hidden states of the reservoir neural network, because of its much higher-dimensional phase space, can possibly live in distinct regions. Using short-term time series of the target attractor as a cue or warming signal to recall it, we can achieve a near 100% success rate of attractor recalling when the length of the warming signal exceeds some critical value, as shown in Fig. 5(A). Before injecting a warming signal, the initial state of the neural network is set to be random, so the success rate should be about 1/6 without the cue. As a cue is applied and its length increases, a relatively abrupt transition to near 100% success rate occurs for all six stored attractors. The remarkable success rate of retrieval suggests that all the six attractors can be trained to successfully coexist in this one high-dimensional dynamical system (i.e., the memory RC) with each of its own separate basin of attraction in the phase space of the neural network hidden state $r$. The finding that various attractors, despite their very different properties—ranging from being periodic or chaotic, with varying periods or maximal Lyapunov exponents, among others - all share nearly identical thresholds for warming length, also suggests that this threshold is more characteristic of the reservoir computer itself rather than the target states.

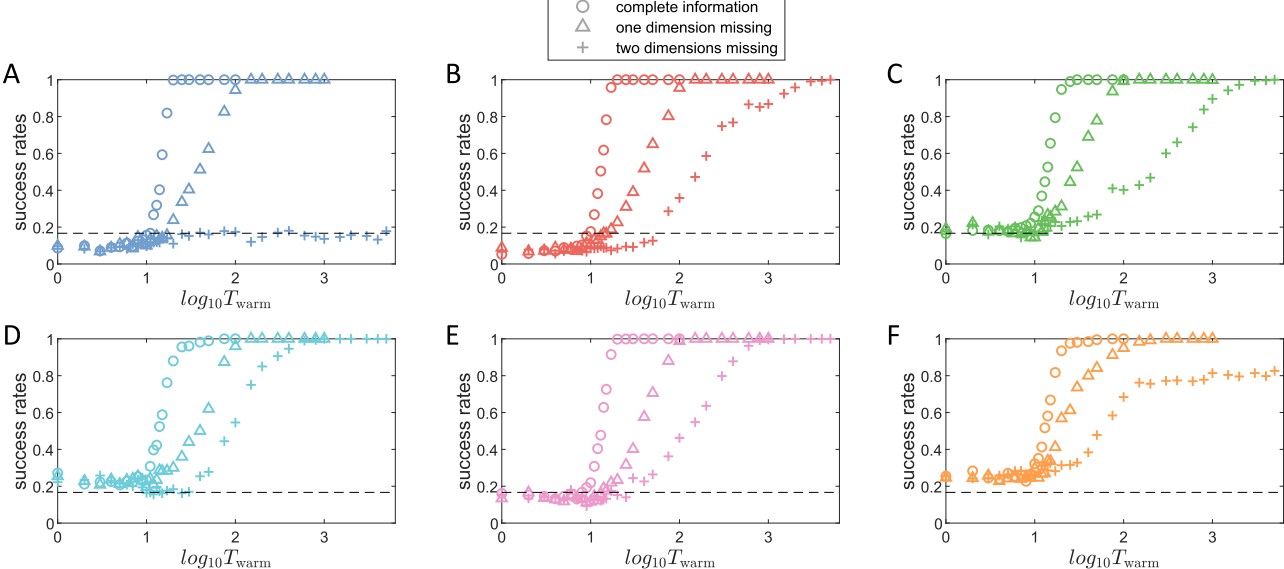

**Fig. 6 | Attractor retrieval with partial cues in index-free reservoir-computing memory. A–F** Shown is the success rate of memory retrieval of the six different chaotic attractors arising from 3D dynamical systems, respectively, with (i) full 3D cues, (ii) 2D cues with one dimension missing, and (iii) 1D cues with two dimensions missing. For 2D cues, a longer warming time is needed to achieve 100% success rate. For 1D cues, in most cases 100% success rate can be achieved with a longer warming time, but it can also occur that the perfect success rate cannot be achieved, e.g., (**A**), (**F**).

It is possible to arbitrarily pick up a "menu" of the attractors and train them to coexist in a memory RC as one single dynamical system. A question is: what are the typical structures of the basins of attraction of the memorized attractors? An example of the 2D projection of the basin structure is shown in Fig. 5(B). In a 2D plane, there are open areas of different colors, corresponding to the basins of different attractors, which are critical for the stability of the stored attractors. More specifically, when the basin of attraction of a memorized attractor contains an open set, it is unlikely for a dynamical trajectory from the attractor to be "kicked" out of the basin by small perturbations, ensuring stability. Figure 5(C) shows the probability (success rate) for a dynamical trajectory of the reservoir network to stay in the basin of a specific attractor after being "kicked" by a random perturbation. For a small perturbation, the probability is approximately one, indicating that the trajectory will always stay near the original (correct) attractor, leading essentially to zero retrieval error in the dynamical memory. Only when the perturbation is sufficiently large will the probability decrease to the value of 1/6, signifying random transition among the attractors and, consequently, failure of the system as a memory. We note that there are quite recent works on basin structures in a high-dimensional Kuramoto model where a large number of attractors coexist[61], which bears similar patterns to the ones observed in our memory RCs. Further investigation is required to study if these patterns can be truly generic across different dynamical systems.

The echo state property of a reservoir computer stipulates that a trajectory from an attractor in its original phase space corresponds to a unique trajectory in the high-dimensional phase space of the RNN in the hidden layer[11,62–64]. That is, the target attractor is be embedded in the RC hidden space. Figure 5(D) shows how the RC state approaches this embedded target state during the warming by the cue. This panel shows the distances (cyan curves) between each of the 100 trajectories of the reservoir neural network and the embedded memorized target attractor versus the cue duration during warming. (The details of how this distance is calculated are given in SI.) The ensemble-averaged distance (the red curve) decreases approximately exponentially with the cue length, indicating that the RC rapidly approaches the memory state's embedding.

The results in Fig. 5 suggest the following picture. The basin structure in Fig. 5(B) indicates that, when a trajectory approaches the target attractor, it usually begins in a riddled-like region that contains points belonging to the basins of different memorized attractors before landing in the open area containing the target attractor. The result in Fig. 5(C) can be interpreted, as follows. The recall success rate when the RC is away from the open areas and still in the riddled-like region is almost purely random (1/K). In an intermediate range of distance where two types of regions are mixed (as the open areas do not appear to have a uniform radius), the recall success rate increases rapidly before entering the pure open region and reaches 100% success rate. This is further verified by Fig. 5(D), where the two horizontal black dashed lines indicate the two distances that equal the magnitudes of perturbation in Fig. 5(C) under which the rapid decline of success rate begins and ends. That is, the interval between these two black dashed lines in Fig. 5(D) is the interval where the RC dynamical state traverses the mixed region. The cue lengths at the two ends of the curve in Fig. 5(D) in this interval are 10 and 50 time steps, which agree well with the transition interval in Fig. 5(A). Taken together, during the warming by the cue, the memory RC begins from the riddled-like region travels through a mixed region and finally reaches the open areas. This journey is reflected in the changes in recall success rate in Fig. 5(A).

It is worth studying if partial information can also successfully recall memorized states in an index-free memory RC. In particular, for the results in Fig. 5(A1–A6), the cue signals used to retrieve any stored chaotic attractor have the full dimensions as that of the original dynamical system that generates the attractor. What if the cues are partial with some missing dimension? For example, if the attractor is from a 3D system, then a full cue signal has three components. If one is missing, the actual input cue signal is 2D. However, since the reservoir network still has three input channels, the missing component can be compensated by the corresponding output component of the memory RC as a feedback loop. (This rewiring scheme was suggested previously[50].) Figure 6 shows, for the six attractors in Fig. 5(A1–A6), the success rate of retrieval versus the cue length or warming signal, for three distinct cases: full 3D cue signals, partial 2D cue signals with one missing dimension, and partial 1D cue signals with two missing dimensions. It can be seen that 100% success rates can still be achieved in most scenarios, albeit longer signals are required. The thresholds of cue length, where the success rate begins to increase significantly

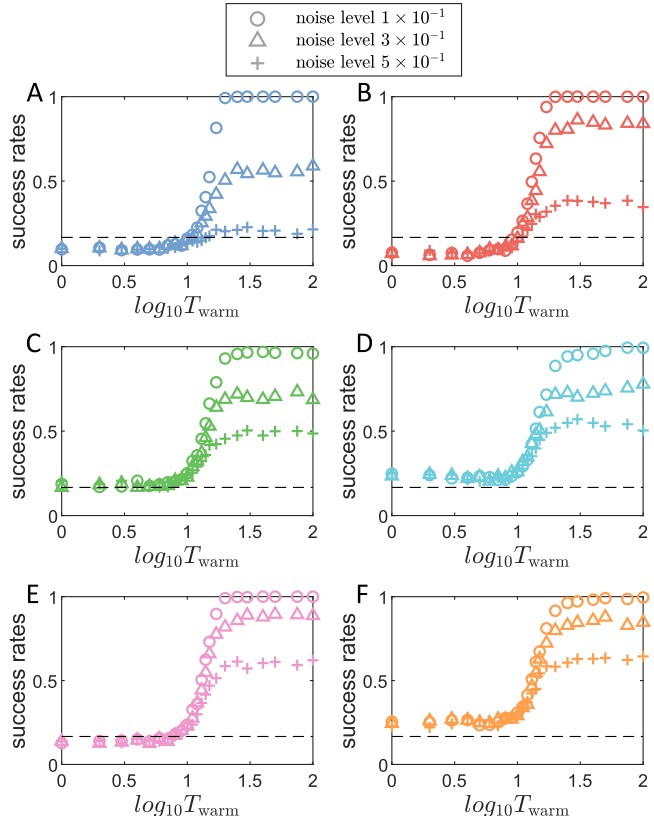

**Fig. 7 | Attractor retrieval with noisy cues from index-free reservoir memories.**
**A**–**F** The success retrieval rate of the six distinct chaotic attractors in Fig. 1(**B**) (from left to right), respectively, versus the cue length.

larger than a random recall (around 1/6), are postponed to a similar degree for all the 2D cases among different attractors. These thresholds are also postponed to another similar degree for all the 1D cases. This result further suggests that the threshold of cue length is a feature of the RC structure and properties rather than a feature of the specific target attractor, with the same way of rewiring feedback loops yielding the same thresholds. (A discussion taking into account all the possible combinations of missed dimensions is provided in SI.)

A further question is: what if the warming signals for the index-free reservoir-computing memory contain random errors or are noisy? Figure 7 shows the retrieval success rate versus the cue length for three different levels of Gaussian white noise in the 3D cue signal. For all six chaotic attractors, a 100% success rate can still be achieved for relatively small noise, but the achievable success rate will decrease as the noise level increases. Different from the scenarios with partial dimensionality, the threshold of required cue length where the success rate jumps significantly from random memory retrieval (1/6) remains the same with different noise levels. We conclude that noisy cues can lower the saturated success rate, while partial information (with rewired feedback loops) can slow down the transition process from random retrieval to saturation in success rate and sometimes lower the saturated success rate.

## Discussion

Traditional neural network models of artificial associative memories, such as the celebrated Hopfield neural networks, are designed to store and retrieve static patterns. To memorize and recall complex dynamical patterns such as chaotic attractors, machine learning based on the reservoir-computing type of RNNs is appropriate because of their ability to produce closed-loop, self-dynamical evolution. In a typical reservoir machine, the intrinsic dynamics are produced by a complex

network of a large number of nonlinear nodes and are high-dimensional. Suppose the attractors to be stored and retrieved, even if they are chaotic, come from relatively low-dimensional dynamical systems. It is then possible for the high-dimensional reservoir system to "accommodate" the attractors in different regions of the phase space. Through appropriate training, each target attractor to be memorized can be made to live in a subregion with its own basin of attraction in the sense that there exists a dynamical invariant set in the subregion which, when recalled, will generate the target low-dimensional attractor, making reservoir-computing based memory possible. The reservoir memory system so designed is dynamic: the intrinsic high-dimensional state vector or the trajectories on the distinct dynamical invariant sets evolve in time continuously. This is the general principle underlying our present work.

The workings of our reservoir-computing-based memory consist of two phases: training for storing the attractors and testing to retrieve any desired attractor. During the training phase, time-series signals from all the attractors to be memorized are used as input to the reservoir neural network to determine certain connection weights in the neural architecture. Successful training serves to place different attractors in different regions of the high-dimensional phase space of the reservoir dynamical network, where each attractor corresponds to a dynamical invariant set, together with its basin of attraction, in some regions of the phase space. After training, the output vector of the reservoir system becomes its input, leaving the system in its own closed-loop dynamical evolution. To recall or retrieve an attractor, either an index-based scheme labeling different attractors or certain cue signals from the attractor to be recalled can be used to drive the dynamical evolution of the reservoir network to output a dynamical trajectory from the invariant set corresponding to the target attractor. The index-based scheme essentially defines a location-addressable memory, while cue-signal-based recalling effectively turns the reservoir-computing system into a content-addressable memory. We demonstrated that, for the location-addressable scenario, the system can memorize a large number of dynamical attractors, including chaotic attractors. Various scaling relations were uncovered between the number of attractors that can be memorized and the size of the reservoir neural network. For the content-addressable scenario, the stored attractors can be recalled using relatively short cue signals even when a certain dimension of the cue signal is missing, and there is noise.

The learning scenario in our work is batch learning with reservoir computing. We have demonstrated that it is computationally efficient and can successfully memorize and recall hundreds of dynamical attractors. The learning scheme is different from classic sequential learning in neuropsychology and in some machine-learning applications, where training is done by one memory state after another[65]. A difficulty with sequential learning is the possible occurrence of catastrophic forgetting[66], where the capability of performing some previously learned tasks can diminish due to changes in the network weights. There were methods for mitigating catastrophic learning[66–69], but developing RNN-based memory for complex dynamical attractors through sequential learning remains to be an open problem.

Itinerancy between attractor states in neural systems is a phenomenon that has attracted much attention[70,71]. As there is multifunctionality in our memory RCs, it is possible that such phenomena can emerge in our systems. There are three different ways of observing itinerancy, which are induced by a fluctuating input signal, transient behaviors, or noises. In Fig. 2(A), we have demonstrated itinerancy among different states under a fluctuating input signal. In an index-free memory RC, we can also observe itinerancy among transient states, with an example shown in SI in Fig. S5. However, how to properly train the memory RC to have the desired itinerancy behavior among transients needs to be investigated. For a real-world neural network, either a biological one or one realized by a physical system (e.g., by analog electronics[72], by a photonic system[73], or by morphological computing

on soft materials[74]), noises on the neurons are usually inevitable. In SI, we show how noises on the neurons in memory reservoir computers can result in spontaneous itinerancy among the attractor states. We hope our results, as dynamical models of itinerancy, can provide useful insights.

The focus of this work is on applying reservoir computing to store and recall a set of countable patterns. In the real brain, a continuum of memory states can exist[41,75]. For attractors arising from a single dynamical system, previous works on adaptable reservoir computing proposed using a continuous bifurcation-parameter channel to anticipate the future dynamical states of the target system[29–33,48], with training based on time series from a small number of parameter values. In Fig. 4(A), the "bifurcation" task sheds light on the application of developing reservoir computers to store and retrieve a continuum of attractors from different dynamical systems.

## Methods

Several different types of dynamical states are used as the target states for the memory RCs to memorize. They include periodic or chaotic attractors generated from numerical integration of corresponding nonlinear ordinary differential equations, periodic trajectories defined in explicit forms as functions of time, and trajectories processed from short videos. In the latter two cases, the original trajectories do not come from dynamical systems, but they can still be trained as attractors in our reservoir computers.

### The six attractors

The six attractors in Fig. 1(B), Fig. 2(A–E), Fig. 3(A), Fig. 5, Fig. 6 and Fig. 7 are: (1) a Lissajous trajectory, (2) a periodic attractor from a Sprott system[53], (3) the classic, chaotic Lorenz attractor, (4) the classic chaotic Rössler attractor (5) a chaotic attractor from a food-chain system[76], and (6) a chaotic attractor from the Hindmarsh-Rose neuron system[77]. The detailed equations and definitions of these memory attractors used in this work are given in SI.

### The 16 Sprott attractors

All 16 chaotic attractors are illustrated in Figs. 1(C) and 2(G, H, I) are generated from the Sprott systems[53]. The time step for numerical integration is $dt = 0.01$, and the time step (temporal resolution) used in the final time series for training and testing is $\Delta t = 0.1$. All three dimensions are memorized by the memory RCs. These data are normalized such that the time series for each dimension has zero mean and unit standard deviation. The time scales of these attractors have a similar order of magnitude, where each oscillation cycle requires 30–60 time steps. If the natural time scales of the attractors to be memorized are drastically different, introducing heterogeneous leakages[78] or time delay[72] into the dynamics of the artificial neuron in the reservoir network can be effective for achieving the training goal.

### Dataset #1

To obtain the scaling law between the number $K$ of attractors that can be memorized and the size $N$ of the reservoir network, hundreds of distinct attractors are needed to ensure at least two orders of magnitude of variation in $K$. The actual number of the attractors in this pool should be larger than the largest number $K$ used to calculate the scaling law to reduce statistical fluctuations caused by some special features of certain attractors. We set out to generate 10,000 distinct attractors. To find such a large pool of distinct attractors is extremely challenging. Our procedure is as follows. We sample 2–5 random points in a 2D plane with 200 time steps and a constant height corresponding to the interval of [−1, 1]. We then fit a fourth-order Fourier series to these random points to obtain a continuous curve with a period of 200-time steps. Each dimension of any attractor is generated independently, and all the 10,000 attractors are also generated independently.

### ALOI videos

This dataset[56] contains 1,000 short videos of rotating objects on a black background. Each video has 72 frames and forms a loop as the object rotates an entire circle back to the initial state. Each frame is a 384 times 288-pixel gray-scale image. We only use the odd number frames as our training and testing data to make our task easier and save computational time. We then perform a dimension reduction based on principal component analysis, as most pixels in the video are just the black background, and most of the remaining pixels are highly correlated. We take the first two principal components as the training and testing data. Therefore, each dynamical state is a two-dimensional periodic state with a period of 36 time steps.

### The "bifurcation" task dataset

All the states are generated by the following dynamical equations:

$$\frac{dR}{dt} = R\left(1 - \frac{R}{K_f}\right) - \frac{x_c y_c CR}{R + R_0}, \tag{1}$$

$$\frac{dC}{dt} = x_c C\left[\frac{y_c R}{R + R_0} - 1\right] - \frac{x_p y_p PC}{C + C_0}, \tag{2}$$

$$\frac{dP}{dt} = x_p P\left(\frac{y_p C}{C + C_0} - 1\right), \tag{3}$$

with $x_c = 0.4$, $y_c = 2.009$, $x_p = 0.08$, $y_p = 2.876$, $R_0 = 0.16129$ and $C_0 = 0.5$. The differences among the states are created by varying the value of $K_f$ in the interval of [0.92, 1], where multiple bifurcations happen, and there are periodic regions and chaotic regions mixed in this interval. A bifurcation diagram of this system in this interval can be found in Fig. 2 of ref. 29. The time step (temporal resolution) of the final time series used as training and testing data is $\Delta t = 1$. All three dimensions are memorized by the memory RC.

## Data availability

The data generated in this study, including both the training time series (as the dynamical states to be memorized) and weights of the reservoir computers, can be found in this OSF repository: https://osf.io/yxm2v/[79].

## Code availability

The codes used in this paper can be found in the repository: https://github.com/lw-kong/Long-Term-Memory-in-RC[80].

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

## Acknowledgements

We thank Dr. J.-J. Jiang for stimulating discussions during the initial phase of the study. We also thank Dr. Andrew Flynn for insightful discussions and comments. This work was supported by the Air Force Office of Scientific Research under Grant No. FA9550-21-1-0438 (to Y.-C.L.). This work was also supported by the Eric and Wendy Schmidt AI in Science Postdoctoral Fellowship, a Schmidt Futures program (to L.-W.K.), and by the U.S. Army Research Institute under Award No. W911NF2310300 (to G.A.B.).

## Author contributions

L.-W. K., G.A.B., and Y.-C. L. designed the research project, the models, and methods. L.-W. K. performed the computations. L.-W. K., G.A.B., and Y.-C. L. analyzed the data. L.-W. K. and Y.-C. L. wrote and edited the manuscript.

## Competing interests

The authors declare no competing interests.
