## [Peer Review File · Nature Communications]

REVIEWER COMMENTS

Reviewer #1 (Remarks to the Author):

The manuscript entitled “Reservoir-computing based associative memory for complex dynamical attractors” proposes setups involving reservoir computing (RC) for models of associative memories for complex dynamical attractors. The authors demonstrate that a single reservoir is able to memorize a large number of periodic and chaotic attractors. They further propose novel control strategies to retrieve arbitrary attractors by starting the trained reservoir with cue signals and to switch between the attractors.

The paper outlines some new and very interesting aspects of reservoir computing that might excel other AI-based retrieval and control mechanisms. The results may also have a significant impact on other fields of research. It is thus potentially worth for publication in nature communications.

Nevertheless, I have some major and minor points that need to be properly addressed before I can recommend the paper for publication.

1) Major points:

a) While it is very impressive to see that a single reservoir is able to memorize a large number of periodic and chaotic attractors, the underlying property of RC of being multifunctional is conceptually not new, but rather already described in some of the articles cited in the manuscript. The presented results are thus “just” an - admittedly impressive - extension of existing methods. Therefore, this part of the paper would not necessarily warrant a publication in nat. comm.

The control methods for retrieval of and switching between attractors are, on the other hand, in my opinion new, very exciting and might have a large application potential ranging from engineering tasks to medicine. The authors should therefore rearrange their manuscript in such a manner that this innovative part of the work is put into the focus.

b) Certainly, the figures of the predicted set of attractors are already convincing. However, a quantitative assessment of the similarity between the true and reconstructed trajectories using appropriate measures is fully missing. Therefore, a quantitative analysis of the prediction results by means of e.g. Lyapunov exponents, correlation dimension together with a corresponding validation (How do e.g. the results depend on varying choice of weights in the reservoir?) is to be added.

c) The description of the methods (also and especially in the supplementary material) is incomplete.

Neither the values of the hyperparameters nor the exact training scheme and hyperparameter optimization are described in detail. The authors should strive to present the methodologies such that their methods can be reprogrammed without reading further papers. With RC being described with only a few equations and training via presumably ridge regression being deterministic a complete and fully transparent description of the methods should be feasible.

2) Minor points:

a) In Fig. 2 the scaling law for the memory capacity is shown. The acceptance rate (50 %) seems to me rather low. How are the results for different acceptance rates? Do the three methods still yield practically the same results? What was the choice of hyperparameters? Did it remain constant for varying K and varying methods?

b) Maybe I missed it, but how is “success” defined in the plots of Figs. 4-6?

c) In Fig 5, one and then two dimensions are left out. Do the results depend on which of the three dimensions are left out? And Why? I guess it could well be the case. It might be insightful and have some relevance for practical applications to do some kind of permutation analysis for some prominent examples (Lorenz, Rössler...).

d) Throughout the manuscript I found careless mistakes (missing words like “the” or “be”) that need to be corrected.

Reviewer #2 (Remarks to the Author):

In this manuscript, the authors train a reservoir computer to reconstruct the dynamics of multiple chaotic attractors. The concept of the work is very interesting. However I cannot recommend that this manuscript be published in its current form. Due to aspects of the work not being adequately

described, I also find it difficult to judge whether the results are significant enough to be published in NCOMM. My concerns about the manuscript are as follows.

1. A thorough description of the reservoir training procedure is critical to understanding what has been done in this manuscript and should therefore be in the methods section of the paper not the supplemental material.
2. If I have interpreted the supplemental material correctly, then the reservoir is trained using one long input sequence composed of trajectories from multiple different chaotic attractors. I have not come across this approach before. If this is a new training method then it should be highlighted, otherwise the appropriate papers should be cited.

I find it very interesting that this approach works.

3. In the graphical depiction of the reservoir in Fig1A it would have helped my understanding if W_{index} was also included.
4. The authors use sentences like “The recalled attractors closely resemble the original attractors”, but do not specify how they determine whether an attractor is successfully recalled or not. The metric used needs to be specified, as this could critically influence the results, especially when many attractors are being recalled.
5. In the “Scaling law for memory capacity” section of the manuscript the authors describe their numerical method for determining the scaling law. This description was not sufficient for me to understand their process. Was a separate RMSE calculated for each attractor for every reservoir? How was the reservoir initialized to test the performance for the different attractors? How was the validation time chosen and how does it relate to the Lyapunov times of the different chaotic attractors? In general, I do not think it is good practice to use a fixed RMSE threshold for dynamical systems with different amplitudes (for example as shown in Fig1B), nor should a fixed validation time be used. The validation times should be related to the timescales of the dynamics of the individual attractors.
6. No parameters for the simulations are specified. The length of the training sequences is particularly important to know.

7. If all the above points are addressed one very important point remains. Based on Fig.2 I see no benefit of the approach presented in this manuscript. According to the scaling law determined by the authors there is no benefit of training one large reservoir to be able to recall many attractors compared with training many small reservoirs to recall one attractor each. Furthermore, in the latter case there would be no issues with switching. I would also expect that the attractors would be reproduced more accurately in the latter case. A proper comparison of the performance in the two cases is essential to determine whether the results of this manuscript could be impactful.

Reviewer #3 (Remarks to the Author):

In "Reservoir-computing based associative memory for complex dynamical systems," the authors demonstrate several strategies for storing and retrieving low-dimensional dynamical systems stored in feedback reservoir computers (RCs). The authors mainly explore two strategies: location-addressable and context-addressable. For the location-addressable RC, the authors add an index channel, and explore several indexing schemes, including a discrete index grid, binary encoding, and one-hot encoding. The authors demonstrate a scaling law spanning roughly two orders of magnitude between the network size and number of stored attractors, with an exponent of roughly 1, and the authors provide a mechanism for how the different schemes lead to increased capacity. Further, the authors demonstrate switching between attractors is not always successful, depends heavily on the destination state, and provide three approaches---switching to intermediate states, a joint memory RC and classifier RC with added noise, and reminder cue signals---to improve the transition success. For the context-addressable memory RC, the authors demonstrate good recall using complete and partial information presented to the RC, and quantify the dependence of attractor basis on small perturbations.

Overall, this manuscript tackles several important questions across and at the intersection between multiple fields. Successful control strategies of recall under noisy perturbations and high-storage capacity are crucial both for the neuroscientific understanding of the brain and the engineering applications of physical reservoir computing. My comments are below.

Major Comments:

1) Motivation and prior work. First of all, I believe this work to have many novel and significant contributions including some of the control strategies for improving recall. I believe it would help properly contextualize the paper if the authors made more explicit which of their results are novel and which are extensions of prior work. For example:

- There are several instances of index-based reservoir-computing memory and transitioning between memories, some even cited by the authors (ref. 37).

- There are several instances of associative recall using warming up via a short recall signal, some even cited by the authors (ref. 54).

2) Lower-dimensional parameterization of the 16 Sprott attractors (ref. 44). In figure 1, the authors demonstrate the storage of 16 Sprott attractors using a 2-dimensional index. Could the success of this learning be due to the RC learning a lower-dimensional analytical parameterization of the quadratic nonlinearities in the generating dynamic functions? A simple way to test this is to randomize the ordering of which attractor goes into each point on the 4x4 grid. This distinction is

significant because it is much easier for an RC to learn an analytical parameterization than 16 distinct dynamical attractors. A more stringent test would be to mix in chaotic attractors from other families, especially those with higher-order polynomials.

3) Power law. I am a bit skeptical about the generalizability of the result in figure 2. Looking at the SI, the 10,000 distinct attractors are most likely generated in an analytically much lower-dimensional manner (i.e. while the periodic attractors might genuinely be distinct from each other, there may be a much lower-dimensional parameterization of these 10,000 attractors that can instead be learned by the RNN). Theoretically, it is known that RCs generate a nonlinear basis of time-history of its inputs, and that feedback serves to use this basis to define recurrent equations (in RC space) that stabilize the dynamics. What surprises me is not that the scaling is almost linear ($\gamma \approx 1$) theoretically, because each new dynamical system below some polynomial order will require a linear number of additional RC units to reconstruct the dynamics. Instead, what surprises me is that the scaling is numerically almost linear, implying that double floating point precision is sufficient to distinguish and even stabilize hundreds of distinct dynamical systems. Some analyses that I would like to see are:

- What is the norm of the readout matrix used for feedback for the largest number of attractors, K (and generally as a function of K)? I would expect the norm would have to be very large to accurately distinguish between so many attractors using the RC's randomly generated dynamical basis.

- What is the reconstruction accuracy in the training of the readout matrix? The reconstruction accuracy would have to be extremely good to accurately reconstruct hundreds of distinct attractors. If the reconstruction accuracy is not very good, then that may suggest the readout matrix is learning to approximate some low-dimensional variety that generates the attractors in the first place.

- Can the authors go into more detail about the generation of the 10,000 distinct attractors, and perhaps try a different method to assess the robustness of Figure 2?

Reviewer #4 (Remarks to the Author):

This paper discusses the development and application of reservoir-computing-based memories for storing and retrieving complex dynamical attractors. Traditional neural network models were limited to static patterns, while reservoir computing enables the handling of complex dynamic patterns, even chaotic attractors. The paper outlines two common recalling scenarios in neuropsychology: location-addressable with an index channel and context-addressable without such a channel.

In the location-addressable retrieval scenario, a single reservoir computing machine can memorize numerous periodic and chaotic attractors, each retrievable with a specific index value. The paper provides control strategies for switching among these attractors and uncovers an algebraic scaling law between the number of stored attractors and the reservoir network's size. In the context-addressable retrieval scenario, the multistability with cue signals is exploited. With the increase of the length of the cue signal, a high success rate can be achieved for attractor retrieval. The paper offers insights into developing long-term memories for complex dynamical patterns. The paper is well-structured, providing a clear introduction and explanation of the topic. However, the embedding multiple low-dimensional attractors in a single reservoir-computing system and achieving switching among them using external input cues has been proposed and explored to some extent in [1]. Thus, the following issues should be addressed.

- (1) The article presents some cases where the switching between attractors failed. Did these failures lead to convergence to other embedded attractors or unknown attractors within the high-dimensional reservoir space, or was it due to the destination attractor not being effectively and stably learned? More explanations are required.
- (2) Intuitively, the level of difficulty in switching attractors is expected to increase with the increase of the number of the embedded attractors. Is there a correlation between the success rate of switching and the quantity of embedded attractors?
- (3) It is interesting that a scaling law between the number of attractors and the network size is uncovered. However, this article only considers the case of 3-dimensional attractors. I am curious to know if the parameter γ of the scaling law is related to the dimensionality or the box-counting dimension of the embedded attractors. Furthermore, I wonder whether or not there is an upper bound on the memory capacity of reservoir-computing to store attractors.
- (4) The article utilizes validation errors over a period of time to assess whether or not the attractors have been successfully memorized. However, the objective of this study is long-term memory rather than short-term memory. Therefore, it might be more appropriate to select indicators that better characterize long-term patterns of the systems, such as the maximum Lyapunov exponent, to determine the success of memory retention.
- (5) The article introduces three effective control strategies to enhance the success rate of attractor switching. However, there is no accompanying data in the study to clearly demonstrate the specific improvement each method brings to the success rate.
- (6) For the feedback control, an asymmetric distance between each pair of stored attractors is proposed, which is associated with the time duration that the random perturbation lasts. However, the article does not explicitly specify the length of the random signals required for switching among attractors, which is a crucial metric for evaluating the effectiveness of this control strategy.
- (7) Is this type of reservoir network able to apply the time-series prediction, such as [2] ?

[1] Lu Z, Bassett D S. Invertible generalized synchronization: A putative mechanism for implicit learning in neural systems. *Chaos: An Interdisciplinary Journal of Nonlinear Science*, 2020, 30(6).

[2] Chen P, et al. Autoreservoir computing for multistep ahead prediction based on the spatiotemporal information transformation. *Nature Communications*, 2020, 11:4568.

Point-by-point response to referee comments

Report of Reviewer #1

General comment: *“The manuscript entitled “Reservoir-computing based associative memory for complex dynamical attractors” proposes setups involving reservoir computing (RC) for models of associative memories for complex dynamical attractors. The authors demonstrate that a single reservoir is able to memorize a large number of periodic and chaotic attractors. They further propose novel control strategies to retrieve arbitrary attractors by starting the trained reservoir with cue signals and to switch between the attractors.*

The paper outlines some new and very interesting aspects of reservoir computing that might excel other AI-based retrieval and control mechanisms. The results may also have a significant impact on other fields of research. It is thus potentially worth for publication in nature communications.

Nevertheless, I have some major and minor points that need to be properly addressed before I can recommend the paper for publication.”

Response: We appreciate that the referee considered that our paper “outlines some new and very interesting aspects of reservoir computing that might excel other AI-based retrieval and control mechanisms”, and that “the results may also have a significant impact on other fields of research”. We also appreciate the other comments of the referee and have made considerable extensions and revisions to our paper accordingly.

Comment 1a: *“While it is very impressive to see that a single reservoir is able to memorize a large number of periodic and chaotic attractors, the underlying property of RC of being multifunctional is conceptually not new, but rather already described in some of the articles cited in the manuscript. The presented results are thus “just” an - admittedly impressive - extension of existing methods. Therefore, this part of the paper would not necessarily warrant a publication in nat. comm. The control methods for retrieval of and switching between attractors are, on the other hand, in my opinion new, very exciting and might have a large application potential ranging from engineering tasks to medicine. The authors should therefore rearrange their manuscript in such a manner that this innovative part of the work is put into the focus.”*

Response: We appreciate that the reviewer considered that “the control methods for retrieval of and switching between attractors” are “very exciting and might have a large application potential ranging from engineering tasks to medicine”, and that the other results with multifunctional RC are “impressive”. In the new version, we have taken the reviewer’s advice to emphasize the control methods for retrieval of and switching between attractors, through a series of changes and rewriting as listed below.

- We have rewritten and substantially extended the relevant sections with many additional results in this switching part. We have also re-plotted or added multiple relevant figures both in the main text and in the SI.
- We have reorganized our paper and moved relevant sections forward in the main text to emphasize them.
- We have put more emphasis on this part in our introduction section.

As a brief summary of the additional results and discussions, we introduced a dynamical understanding of the difference between successful and failed switching and provided a clearer explanation of why the success rate is more correlated to the destination states. Based on this useful understanding, we qualitatively evaluate the

potential of our different methods to achieve a better success rate. Furthermore, we added quantitative evaluation of the increases in recalling or switching success rates with the aid of each of our control methods to quantify these methods' performance.

Comment 1b: *“Certainly, the figures of the predicted set of attractors are already convincing. However, a quantitative assessment of the similarity between the true and reconstructed trajectories using appropriate measures is fully missing. Therefore, a quantitative analysis of the prediction results by means of e.g. Lyapunov exponents, correlation dimension together with a corresponding validation (How do e.g. the results depend on varying choice of weights in the reservoir?) is to be added.”*

Response: We appreciate the reviewer's advice and agree that using these indicators to measure the long-term fidelity of attractor reconstruction can provide strong support to our approach to multifunctional RC. We have added results with the maximum Lyapunov exponents and correlation dimensions in the main text and the SI. These new results can justify that the long-term features of the memorized chaotic state (the “climate”) can be faithfully reproduced during memory retrieval. In the main text, we added the following discussion:

- We evaluate the fidelity of the recalled states in the long term by calculating the maximum Lyapunov exponents and correlation dimension. We train an ensemble of 100 different memory RCs, recall each of the memory states, and generate outputs continuously for 200,000 steps. The two measures are calculated within this generation process and compared with the ground truth. As an example, for the Lorenz attractor, with a ground truth correlation dimension of 2.05 ± 0.01 , in 96% of the memory RCs, we retrieve an attractor with a correlation dimension within 2.05 ± 0.02 . As for the maximum Lyapunov exponent, the ground truth for the Lorenz attractor is about 0.906, and 91% of the memory RCs fall in the small range of 0.906 ± 0.015 . A more comprehensive result is shown in the SI, with similarly high fidelity for all other states. This result shows that it is indeed the long-term “climate” that the memory RC has learned and can reproduce.

In Supplementary Note 2, we further provide results of the maximum Lyapunov exponents of the reconstructed chaotic attractors, and demonstrate how this performance varies across different levels of training noise σ_{noise} . The results are demonstrated in Fig. S3, showing that within an optimal noise interval, the maximum Lyapunov exponents of the reconstructed chaotic attractors agree well with the ground truth values.

Comment 1c: *“The description of the methods (also and especially in the supplementary material) is incomplete. Neither the values of the hyperparameters nor the exact training scheme and hyperparameter optimization are described in detail. The authors should strive to present the methodologies such that their methods can be reprogrammed without reading further papers. With RC being described with only a few equations and training via presumably ridge regression being deterministic a complete and fully transparent description of the methods should be feasible.”*

Response: We appreciate the reviewer's advice and have rewritten and substantially extended the method descriptions (both in the Methods section of the main text and in the SI) to provide much more detail to help readers understand and implement. We have also added a section at the end of the main text where we list all the hyperparameters of the reservoir computers for each of the tasks we demonstrate in this paper. In addition, in multiple places throughout the paper, we have provided the values of the relevant parameters used in training or testing.

Comment 2a: “In Fig. 2 the scaling law for the memory capacity is shown. The acceptance rate (50 %) seems to me rather low. How are the results for different acceptance rates? Do the three methods still yield practically the same results? What was the choice of hyperparameters? Did it remain constant for varying K and varying methods?”

Response: We have added scaling results with a higher acceptance rate (80 %) to the SI, as shown in Fig. S10, where our conclusions are not changed under this variation. We have chosen the acceptance rate as 50% in the main text because this is the value that would result in the least error. If this percentage is set to be closer to the two extremes 0% or 100%, a small fluctuation in the simulation results can become a much larger relative error in the success rate curve around the threshold. And this fluctuation around the threshold can result in a larger error in the determined N_C . This issue becomes particularly severe when K is large, where much computational effort would be required to maintain the same level of accuracy in N_C as the 50% threshold case. Besides Fig. S10 in the SI, we have also added the following brief discussion in the main text:

- Here we use 50% as a threshold to minimize the potential error in N_C due to random fluctuations. In the SI, we provide results with 80% as the success rate threshold. Our main conclusions remain the same under this change.

Comment 2b: “Maybe I missed it, but how is “success” defined in the plots of Figs. 4-6?”

Response: For an index-free version of RC-based memory models (with the results shown in Figs. 4-6 in the previous version), determining if a recall is successful is a highly nontrivial task and can be pretty complicated. We should not use a simple short-term performance measurement like root-mean-square error (RMSE) or prediction horizon. Part of the reason is that we usually do not have access to an accurate “ground truth” of a specific trajectory of the corresponding attractor, especially for the chaotic states, so there is no way of directly comparing the recalled trajectory with this “ground truth” trajectory. Furthermore, an evaluation based on calculating maximum Lyapunov exponents or correlation dimensions may also not be the best solution, as two attractors with a similar maximum Lyapunov exponent may actually have different shapes. With features of just a few numbers, we cannot determine if a recall is really accurate or not. Moreover, in many of the tasks we test, we do not necessarily require a persistent reconstruction of the original attractor that can last for thousands of oscillations. In many realistic scenarios, just persisting the memory accurately for dozens of periods can be sufficient and can fit various application scenarios. Thus, neither short-term measures like RMSE or prediction horizon nor long-term measures like Lyapunov exponents or correlation dimensions fit our needs.

To tackle this difficulty, we train another neural network to perform this short-term classification task to determine which memorized states are recalled or if no successful recall is made at all. We train another reservoir computer for this task. As discussed in the SI, we have tested its performance on more than a thousand trials, and it reaches a surprisingly high accuracy, with only 6 trials out of 1,200 trials of deviation from a human labeler, even the 6 trials are intrinsically ambiguous to classify. This neural network, which we call the classifier RC in the paper, can automatically perform the classification.

We have added the following discussion in the text accordingly to make the above points clear to the readers (on pages 10-11):

- While the possibility of index-free memory RC has been pointed out previously [50], as discussed in Introduction, we still lack a quantitative analysis of the mechanism of the retrieval. Such an analysis, in our view, should include an investigation into how the cue signals are affecting the success rate, what

the basin structure of the reservoir computer is, and how to have a dynamical understanding that can connect these two issues. A difficulty with such quantitative analysis is how to efficiently and accurately determine, from a short segment of RC output time series, which memory state is recalled and if any target memory has been recalled at all. We should not use a simple short-term performance measurement like root-mean-square error (RMSE) or prediction horizon. The reason is that we usually do not have access to an accurate "ground truth" of a specific trajectory of the corresponding attractor, especially for the chaotic states, so there is no way of directly comparing the recalled trajectory with this "ground truth" trajectory. Moreover, an evaluation based on calculating maximum Lyapunov exponents or correlation dimensions may also not be the best solution, as both measures require a substantially long persist time series, such as thousands of oscillations. Furthermore, in many realistic scenarios, just persisting the memory accurately for dozens of periods can be sufficient and can fit various application scenarios.

To tackle this difficulty, we train another neural network to perform this short-term classification task to determine which memorized states are recalled or if no successful recall is made at all. To make things simpler, we train another reservoir computer for this task, which we call the classifier RC. We find that this classifier RC can automatically perform the classification tasks efficiently and with surprisingly high accuracy. As discussed in the SI, we have tested its performance on more than a thousand trials, and it reaches a surprisingly high accuracy, with only 6 trials out of 1,200 trials of deviation from a human labeler. And even that 6 trials are intrinsically ambiguous to classify. This classifier RC is also used in the control scheme for index-based memory RC, as shown in Fig. 3(D), as part of the controller.

Comment 2c: *"In Fig 5, one and then two dimensions are left out. Do the results depend on which of the three dimensions are left out? And Why? I guess it could well be the case. It might be insightful and have some relevance for practical applications to do some kind of permutation analysis for some prominent examples (Lorenz, Roessler ...)."*

Response: We have added new results on all the possible combinations of missing dimensions with all the six attractors we were focusing on. They are included in Supplementary Note. 7, as the following, as well as in Fig. S13 and Fig. S14.

- In the main text, we demonstrate how our index-free memory RC can still function and recall target states with partial cues missing some dimensions. However, we only show one missing-dimension scenario for each dimensionality in the main text, while there are more possible combinations of missing dimensions in the three-dimensional target states we test. It could be interesting to see how the retrieval performance depends on the specific combinations of dimensions that are missing. Here, we demonstrate comprehensive results on all the possibilities of missing dimensions with all the six attractors we test. The results are shown in Fig. S13 and Fig. S14. Comparing the six attractors, we observe that the thresholds of the cue length where the success rate begins to rise significantly larger than a random recalling are postponed similarly to the results shown in Fig. 6 in the main text. This again verifies that the threshold of cue length is a feature of the RC structure and properties rather than a feature of the specific target attractor, with the same way of rewiring feedback loops yielding the same thresholds. A decrease in the saturated success rate is also observed in multiple cases, and is more frequent for the 1D cues than the 2D cues, as more information is lost. Among the four chaotic attractors, the Lorenz attractor (in panel (C)) can always reach a 100% success rate. Both the chaotic Rossler system and the chaotic food chain system suffer no decrease in the saturated success rate except in the sole case where only the third dimension

is lost. The most intriguing case is with the periodic Sprott system, where missing the second dimension alone would result in a worse saturated success rate than missing two dimensions. This suggests that having more dimensions hidden during the retrieval does not always make the success rate worse; the relationship between missing dimensions and the change in success rate is much more complicated and system-dependent. Further research is necessary to unveil a possible generic understanding of these interesting phenomena.

Comment 2d: *“Throughout the manuscript I found careless mistakes (missing words like “the” or “be”) that need to be corrected.*

Response: We sincerely apologize for the English errors in the paper. We have fixed all the typos and errors that we can find, and have double-checked the text.

Report of Reviewer #2

General Comments: *“In this manuscript, the authors train a reservoir computer to reconstruct the dynamics of multiple chaotic attractors. The concept of the work is very interesting. However I cannot recommend that this manuscript be published in its current form. Due to aspects of the work not being adequately described, I also find it difficult to judge whether the results are significant enough to be published in NCOMM. My concerns about the manuscript are as follows.”*

Response: We appreciate that the reviewer considered that “the concept of the work is very interesting”. We have revised our paper considerably in response to the referees’ comments.

Comment 1: *“A thorough description of the reservoir training procedure is critical to understanding what has been done in this manuscript and should therefore be in the methods section of the paper not the supplemental material.”*

Response: We have substantially extended the Methods section in the main text to help the readers understand and possibly reproduce our results with no need for further references. We have also added information on the hyperparameters and other training and testing setting configurations, including the training length, to our new version of the paper. They are summarized at the end of the Methods section and Table 1 in the main text.

Comment 2: *“If I have interpreted the supplemental material correctly, then the reservoir is trained using one long input sequence composed of trajectories from multiple different chaotic attractors. I have not come across this approach before. If this is a new training method then it should be highlighted, otherwise the appropriate papers should be cited. I find it very interesting that this approach works.”*

Response: It is indeed interesting that this simple and straightforward approach works. This method has been applied by several previous works. We have added a discussion on this approach in the Methods section in the main paper, including several citations to the previous works that we found which used this approach (on pages 14-15):

- Compared with a more standard approach where only one attractor/state is trained, the trick with our approach (for both indexed and index-free schemes) is that we then concatenate the records of the hidden states $r(\hat{t})$ from different target states together in the temporal dimension to form one (potentially very long) time series $R(\hat{t})$. The training target is processed in the same way, where $v(\hat{t})$ from different states are concatenated in the temporal dimension to form one (also potentially very long) time series $V(\hat{t})$. This “concatenating in time” scheme has been used in several previous work in the context of reservoir computing [29,32,33] when more than one state/attractor is trained. It is pretty interesting that this simple method works at all, especially given that the different states it concatenates can have very different dynamical features, from simple periodic oscillations to various chaotic trajectories with different forms of nonlinearities. The high dimensionality of the RC system and the simplicity of linear regression allow this way of merging different dynamics into one recurrent neural network. This is particularly helpful for our tasks with multiple states to memorize as no forgetting issue would arise.

Comment 3: *“In the graphical depiction of the reservoir in Fig1A it would have helped my understanding if W index was also included.”*

Response: We have included w_{index} in Fig. 1A as suggested. We have also added the following sentence in the description of the indexed memory RC in the main text (on page 3):

- This index value p is modulating the dynamics of the RC network through an index input matrix w_{index} in the input layer, which projects p to the entire hidden layer with weights defined by the entries in W_{index} .

Comment 4: “The authors use sentences like “The recalled attractors closely resemble the original attractors,” but do not specify how they determine whether an attractor is successfully recalled or not. The metric used needs to be specified, as this could critically influence the results, especially when many attractors are being recalled.”

Response: Throughout the paper, we have applied multiple performance measures (and cross-verifications among these measures) to validate the accuracy when we determine if a specific attractor is successfully recalled or not. The methods we use include comparing the maximum Lyapunov exponents between the RC-reconstructed and original attractors, comparing the correlation dimension between the reconstructed and the original, and calculating short-term fidelity by RMSE, prediction horizon, and oscillating regions. Furthermore, a classifier reservoir computer is also used to classify generated trajectories automatically. The following discussions are added to the main paper to make it more clear to the readers of these validation methods:

For the results with six attractors (in the left column of page 4):

- We evaluate the fidelity of the recalled states in the long term by calculating the maximum Lyapunov exponents and correlation dimension. We train an ensemble of 100 different memory RCs, recall each of the memory states, and generate outputs continuously for 200,000 steps. The two measures are calculated within this generation process and compared with the ground truth. As an example, for the Lorenz attractor, with a ground truth correlation dimension of 2.05 ± 0.01 , in 96% of the memory RCs, we retrieve an attractor with a correlation dimension within 2.05 ± 0.02 . As for the maximum Lyapunov exponent, the ground truth for the Lorenz attractor is about 0.906, and 91% of the memory RCs fall in the small range of 0.906 ± 0.015 . A more comprehensive result is shown in the SI, with similarly high fidelity for all other states. This remarkable result shows that it is indeed the long-term “climate” that the memory RC has learned and can reproduce.

For the results on the scaling behaviors (in the left column of page 8):

- We use three different types of validation performance measures to make our results more generic. We use a prediction horizon which is defined by the prediction length before the deviation of the prediction time series reaches 10% of the oscillation amplitude of the real state. This oscillation amplitude is defined by half of the distance of the maximum value minus the minimum value in a sufficiently long time series of that target state. The oscillation amplitude and corresponding deviation rate are calculated for each dimension of the target state, and we take the shortest prediction horizon among all the dimensions of the target state. The prediction horizon is rescaled by the length of the period of the target system to make it comparable among different states. For a chaotic state, we use an average period defined as the average shortest time between two local maximums in a sufficiently long time series of that target state. The threshold of this prediction horizon for a successful recall is set to be two periods of each memory state. The second performance measure we use is a mean square root error (RMSE) with a validation time of four periods (or average periods). The third validation performance measure is defined by the time before the predicted trajectories go beyond a rectangular-shaped region surrounding the real memory state so

that it goes to an undesired region in the phase space. This rectangular-shaped region is set to be 10% larger than the rectangular region defined by the maximum and minimum values of the memory state in each dimension. For these measures, the thresholds for a successful recall can be different across datasets but are always fixed within the same task (the same scaling curve). The thresholds for the prediction-horizon-based measures and region-based measures are always several periods (or average periods for chaotic states) of the target state.

For the results on the index-free memory RCs (in the left column of page 11):

- To tackle this difficulty, we train another neural network to perform this short-term classification task to determine which memorized states are recalled or if no successful recall is made at all. We train another reservoir computer for this task, which we call the classifier RC. We find that this classifier RC can automatically perform the classification tasks efficiently and with surprisingly high accuracy. As discussed in the SI, we have tested its performance on more than a thousand trials, and it reaches a surprisingly high accuracy, with only 6 trials out of 1,200 trials of deviation from a human labeler. And even that 6 trials are intrinsically ambiguous to classify. This classifier RC is also used in the control scheme for index-based memory RC, as shown in Fig. 3(D), as part of the controller.

Comment 5: *“In the ‘Scaling law for memory capacity’ section of the manuscript the authors describe their numerical method for determining the scaling law. This description was not sufficient for me to understand their process. Was a separate RMSE calculated for each attractor for every reservoir? How was the reservoir initialized to test the performance for the different attractors? How was the validation time chosen and how does it relate to the Lyapunov times of the different chaotic attractors? In general, I do not think it is good practice to use a fixed RMSE threshold for dynamical systems with different amplitudes (for example as shown in Fig1B), nor should a fixed validation time be used. The validation times should be related to the timescales of the dynamics of the individual attractors.”*

Response: Inspired by the referee’s comment, we have now introduced three different types of performance measures, as discussed in the following paragraph in the new version of the main paper (on pages 8-9):

- We use three different types of validation performance measures to make our results more generic. We use a prediction horizon which is defined by the prediction length before the deviation of the prediction time series reaches 10% of the oscillation amplitude of the real state. This oscillation amplitude is defined by half of the distance of the maximum value minus the minimum value in a sufficiently long time series of that target state. The oscillation amplitude and corresponding deviation rate are calculated for each dimension of the target state, and we take the shortest prediction horizon among all the dimensions of the target state. The prediction horizon is rescaled by the length of the period of the target system to make it comparable among different states. For a chaotic state, we use an average period defined as the average shortest time between two local maximums in a sufficiently long time series of that target state. The threshold of this prediction horizon for a successful recall is set to be two periods of each memory state. The second performance measure we use is a mean square root error (RMSE) with a validation time of four periods (or average periods). The third validation performance measure is defined by the time before the predicted trajectories go beyond a rectangular-shaped region surrounding the real memory state so that it goes to an undesired region in the phase space. This rectangular-shaped region is set to be 10% larger than the rectangular region defined by the maximum and minimum values of the memory state in

each dimension. For these measures, the thresholds for a successful recall can be different across datasets but are always fixed within the same task (the same scaling curve). The thresholds for the prediction-horizon-based measures and region-based measures are always several periods (or average periods for chaotic states) of the target state.

Our performance measures are now related to the timescales of the dynamics of the individual attractors. For the chaotic attractors, we use the average oscillation period instead of the maximum Lyapunov exponents because we focus more on the critical size N_C of memory RCs, where the memory RC usually does not perform perfectly well. The deviation of the predicted trajectories caused by chaos should not be the dominant factor, as the accumulated errors in the RC iterations usually deviate the trajectories away before a Lyapunov time.

We further demonstrate that these three different measures yield similar scaling results. As shown in Fig. 4(C), the differences in performance measures only result in a constant factor in the scaling behaviors for the examples we show. We have also added the following discussion in the main paper (in the left column of page 10):

- To ensure that our results are generic regardless of the validation measures, in Fig. 4(C), we compare the scaling behaviors of the same task (the “one-hot coding” task) with the three different measures we have. All the three curves have almost the same scaling behavior, except a minor difference in a constant factor.

Comment 6: “No parameters for the simulations are specified. The length of the training sequences is particularly important to know.”

Response: We have added the information on the hyperparameters and other training and testing setting configurations, including the training length, to our new version of the paper. They are summarized at the end of the Methods section and Table 1 in the main text.

Comment 7: “If all the above points are addressed one very important point remains. Based on Fig.2 I see no benefit of the approach presented in this manuscript. According to the scaling law determined by the authors there is no benefit of training one large reservoir to be able to recall many attractors compared with training many small reservoirs to recall one attractor each. Furthermore, in the latter case there would be no issues with switching. I would also expect that the attractors would be reproduced more accurately in the latter case. A proper comparison of the performance in the two cases is essential to determine whether the results of this manuscript could be impactful.”

Response: First, we would like to point out that the imperfect switching success rate issue is essentially an initial condition problem during switching, and it is a generic issue as long as one is learning the dynamic rules of the target system. Even if we have a group of separate networks, each trained with one state, the switching among them would still face the problem of how to initialize them during such switching properly. Therefore, almost all the most important results presented in this paper, including (1) the imperfect switching problem and the dynamical understanding of it, (2) control strategies for better initializing/switching, and (3) all of our results for the index-free content-addressable scenario where all the states have to be coexisting in the same dynamical system, remain useful and are not affected by the comparison between the index-based approach and the separate network approach we are discussing here.

Even if we go with the general idea of having a group of networks for different memorized states, a better approach would be to fix the input and recurrent matrices and just train different output matrices for different

states. In all of our results, the input and recurrent matrices are generated randomly and are not sensitive to the state in which they are targeting. Thus, it is indeed possible that we can have a shared reservoir (shared input and hidden layers) with different readout matrices for different memorized states. This approach is going to require an additional mechanism of choosing different output matrices on the state (which is again going to require an index value as an additional input), unlike our original case where the entire RC network is one integrated neural network, and one never needs to change anything during the entire procedure of retrieval and switching. Still, it is true that this approach can reduce the computational costs. We have added results with this separate W_{out} approach in Fig. 4(A). Moreover, as the hidden layer is shared among different target states, the hidden state r is also shared and persists in the RC as one switches the index value. Therefore, the similar phenomenon of failed switching will still happen in a similar way, and the control strategies we develop can still be applied.

Furthermore, our method has a strong advantage in certain scenarios. In general, one integrated neural network should be more efficient than many separate networks with efficient training methods, as different memorized states can share weights to compress the overall size of the neural network. This is exactly the case of our “bifurcation” task shown in Fig. 4(A). In this example, all the target states come from the same set of dynamical equations but with different parameter values in the equations. The memory RC can utilize such correlations to learn a global representation of the structure among different states, and significantly reduce the computational costs. As shown in Fig. 4(A), the scaling behavior of index-based memory RC on this task is much slower than a linear law. As this research field advances in the future, we expect better training methods that can utilize this type of shared dynamic features in a broader scope of scenarios.

Besides the results shown in Fig. 4(A), we have added multiple discussions in the main text, including the following two paragraphs:

In the left column of page 9:

- The “bifurcation” task is an example to show the potential of our index-based approach, where N_c grows much slower than a linear scale with respect to K , and can even decrease in certain cases. It consists of 100 dynamical states gathered from the same chaotic food chain system but with different parameter values. We basically sweep an interval in the bifurcation diagram of the food chain model, with many bifurcations happening and with both periodic and chaotic states with different periods or average periods. For the index coding, we use a one-dimensional index and assign the states by sorting the system parameter values used to generate these states. The index values are always in the interval $[-2.5, 2.5]$, and are evenly separated on this interval. The fact that we reach a scaling behavior much slower than linear law suggests that when the target states are somehow correlated, and the index values are assigned in a “meaningful” way, the reservoir computer can utilize the similarities among target states for more efficient learning and storing. Moreover, we notice that N_c can even decrease in a certain interval of K . Our understanding is that the amount of total training data increases with larger K , which makes it easier for the reservoir computer to reach a similar performance with smaller N_c . In our other tasks with other training sets, training data from different target states are independent of each other and thus do not experience the aid of training data from other target states. In summary, training with our index-based approach on this dataset is more resource-efficient than having a series of separate reservoir computers trained with each target state and adding another selection mechanism to switch the reservoir computer while operating switching or itinerary behaviors.

In the paragraph from page 9 to 10:

- One of the most important features of reservoir computing is that the input layer and the recurrent hidden layer are generated randomly and, thus, are essentially independent of the target state. One way of

utilizing this convenient feature is to just use the same input and hidden layer for all the target states but with a separate output layer trained for each target state. In such an approach, an additional mechanism of selecting the correct output layer (i.e., W_{out}) during retrieval is necessary, unlike the other approaches in this paper where we always have the same integrated reservoir computer for all the different target states. However, training with separate W_{out} does appear to lower the computational complexity in most cases. Comparing the index-based approach that we have focused on, its efficiency can be gone if the target states are somewhat similar, like in the “bifurcation” task. In this task, training with separate W_{out} still makes the model complexity grow a bit slower than linearly, as we need the same number of independent W_{out} as the number of K , and N_c is still increasing with a power law as shown in Fig. 4(A). However, with the index-based approach, everything grows much slower than linear and is very efficient. Furthermore, as the input layer and hidden layer are shared by all the different target states, the issue of failed switching still exists, and the control methods we have discussed are still useful.

In summary, comparing this separate W_{out} approach with our index-based approach, we conclude that the index-based approach has the following advantages:

1. The entire RC network is one integrated neural network, and one never needs to change any network structure during the entire procedure of retrieval and switching. No additional mechanism for choosing different output matrices is required.
2. The index-based approach can be computationally more efficient when the states are dynamically correlated.

The separate W_{out} approach has the advantage of computational cost when the target states are dynamically uncorrelated or the correlation is not utilized. Both approaches face similar failed switching problems where similar control strategies that we discuss in the main text can be applied.

Report of Reviewer #3

General comment: *“In ”Reservoir-computing based associative memory for complex dynamical systems,” the authors demonstrate several strategies for storing and retrieving low-dimensional dynamical systems stored in feedback reservoir computers (RCs). The authors mainly explore two strategies: location-addressable and context-addressable. For the location-addressable RC, the authors add an index channel, and explore several indexing schemes, including a discrete index grid, binary encoding, and one-hot encoding. The authors demonstrate a scaling law spanning roughly two orders of magnitude between the network size and number of stored attractors, with an exponent of roughly 1, and the authors provide a mechanism for how the different schemes lead to increased capacity. Further, the authors demonstrate switching between attractors is not always successful, depends heavily on the destination state, and provide three approaches—switching to intermediate states, a joint memory RC and classifier RC with added noise, and reminder cue signals—to improve the transition success. For the context-addressable memory RC, the authors demonstrate good recall using complete and partial information presented to the RC, and quantify the dependence of attractor basis on small perturbations.*

Overall, this manuscript tackles several important questions across and at the intersection between multiple fields. Successful control strategies of recall under noisy perturbations and high-storage capacity are crucial both for the neuroscientific understanding of the brain and the engineering applications of physical reservoir computing. My comments are below.”

Response: We appreciate that the referee considered that our paper “tackles several important questions across and at the intersection between multiple fields”, and that “successful control strategies of recall under noisy perturbations and high-storage capacity are crucial both for the neuroscientific understanding of the brain and the engineering applications of physical reservoir computing”. We also appreciate the other comments of the referee and have made considerable extensions and revisions to our paper accordingly.

Comment 1: *“Motivation and prior work. First of all, I believe this work to have many novel and significant contributions including some of the control strategies for improving recall. I believe it would help properly contextualize the paper if the authors made more explicit which of their results are novel and which are extensions of prior work. For example: - There are several instances of index-based reservoir-computing memory and transitioning between memories, some even cited by the authors (ref. 37). - There are several instances of associative recall using warming up via a short recall signal, some even cited by the authors (ref. 54).”*

Response: We appreciate the advice of the reviewer and have made major revisions to the introduction section, as well as many expressions in other sections of the paper. We rewrote the paragraph on prior work as follows (in the left column of page 2):

- A neural network capable of memorizing and retrieving multiple memory states is also an example of a multifunctional neural network. When different states are being recalled, the neural network demonstrates distinct dynamical behaviors. The idea of multifunctional neural networks, and more specifically, multifunctional recurrent neural networks, has been discussed previously with different forms of implementations. One approach is to use an index value to modulate the functionality of the neural networks and store different states with each index value [29, 31, 32, 47, 48]. (In our paper, this approach is called the index-based approach.) However, the problem of recalling has not yet been addressed, which is, of course, a crucial part of associative memory study. Furthermore, when a memory state is difficult to recall, it is important to discuss strategies to help the recalling process efficiently, including some control

schemes. Moreover, no discussions on scaling laws of the memory capacity are available in this context of storing different states in recurrent neural networks. Another approach is to utilize the multistability in the neural networks [49–52], where multiple attractors coexist in the extremely high-dimensional hidden space of the reservoir computer. (In our paper, this approach is called the index-free approach.) Ref. [49] focused on storing fixed points, while Ref. [50–52] explored the possibility of storing two chaotic or periodic states. However, a quantitative study on the dynamical mechanism of how these coexisting dynamical states are retrieved, what the basin structure is like, and how these two questions are connected are all missing but are crucial for our understanding of the memorizing and recall processes. Such a quantitative study becomes possible with our machine-learning-based classifier, which can automatically distinguish among different recalled states and between a successful and failed recall, all with high accuracy.

Some other changes of expressions in the paper include:

In the left column of page 10:

- ... the possibility of index-free memory RC has been shown previously [50],

... In the left column of page 12:

- (We notice that this rewiring scheme was also discussed in Ref. [50].)

Comment 2: “Lower-dimensional parameterization of the 16 Sprott attractors (ref. 44). In figure 1, the authors demonstrate the storage of 16 Sprott attractors using a 2-dimensional index. Could the success of this learning be due to the RC learning a lower-dimensional analytical parameterization of the quadratic nonlinearities in the generating dynamic functions? A simple way to test this is to randomize the ordering of which attractor goes into each point on the 4x4 grid. This distinction is significant because it is much easier for an RC to learn an analytical parameterization than 16 distinct dynamical attractors. A more stringent test would be to mix in chaotic attractors from other families, especially those with higher-order polynomials.”

Response: This is indeed an interesting question that can provide crucial insights into the working mechanism of reservoir computers, and it is also a question we asked ourselves when writing the paper. We have added results by randomly ordering the 16 Sprott attractors in the Supporting Information, in Supplementary Note 3 and Fig. S3. We show 100 different RC networks with the 16 attractors randomly assigned to the index values on the 4×4 lattice. We show that they can all perform well for almost all the attractors, regardless of how the index values are assigned. This result suggests that a lower-dimensional parameterization, which could be computationally beneficial when we want it (as demonstrated in the task “bifurcation” in Fig. 4(A) in the main text), is not a necessary condition for our approach to function.

We would also like to point out that in the first example that we show with six attractors, we made this selection of attractors while considering taking a mix with different families of nonlinear systems. Besides the classic Lorenz and Rossler systems, where the ODEs have quadratic nonlinear terms, we deliberately include the Hindmarsh-Rose (HR) neuron system, where there is a cubic term, and we deliberately include the chaotic food chain system, which has fractional terms.

Comment 3: “Power law. I am a bit skeptical about the generalizability of the result in figure 2. Looking at the SI, the 10,000 distinct attractors are most likely generated in an analytically much lower-dimensional manner (i.e. while the periodic attractors might genuinely be distinct from each other, there may be a much lower-dimensional parameterization of these 10,000 attractors that can instead be learned by the RNN). Theoretically,

it is known that RCs generate a nonlinear basis of time-history of its inputs, and that feedback serves to use this basis to define recurrent equations (in RC space) that stabilize the dynamics. What surprises me is not that the scaling is almost linear ($\gamma \approx 1$) theoretically, because each new dynamical system below some polynomial order will require a linear number of additional RC units to reconstruct the dynamics. Instead, what surprises me is that the scaling is numerically almost linear, implying that double floating point precision is sufficient to distinguish and even stabilize hundreds of distinct dynamical systems. Some analyses that I would like to see are:

- What is the norm of the readout matrix used for feedback for the largest number of attractors, K (and generally as a function of K)? I would expect the norm would have to be very large to accurately distinguish between so many attractors using the RC's randomly generated dynamical basis.

- What is the reconstruction accuracy in the training of the readout matrix? The reconstruction accuracy would have to be extremely good to accurately reconstruct hundreds of distinct attractors. If the reconstruction accuracy is not very good, then that may suggest the readout matrix is learning to approximate some low-dimensional variety that generates the attractors in the first place.

- Can the authors go into more detail about the generation of the 10,000 distinct attractors, and perhaps try a different method to assess the robustness of Figure 2?"

Response: Inspired by the referee's advice, we have substantially extended our scaling law section with multiple new sets of attractors and different training and recalling schemes, leading to a series of new results and discussions. We find that scaling behavior is very complicated and can vary among different scenarios. These results are shown in Fig. 4 of our new version of the paper. We indeed find a variety of different scaling behaviors other than a linear law. However, this variety also unveils the previously hidden complexity of the phenomenon. We believe it should be very difficult to draw a universal relationship between the features of the attractors and the scaling law. As an example, we find that for index-free memory RCs, factors such as the degree of overlap among the target states can significantly affect scaling behavior.

In Fig. S11, we provide the reconstruction errors and 2-norm of the output matrices w_{out} of our index-based memory RCs trained with $K = 16$, $K = 64$, and $K = 256$ attractors. All the RCs are around the critical network size N_c . The reconstruction error is calculated by the RMSE on the training data of all target states after training. We find that around N_c , the reconstruction error around N_c is always at the level of 10^{-3} , which is indeed very small and gets smaller with a larger K . The 2-norm of w_{out} is also increasing with an increasing K , but it actually increases very slowly and is close to linear. We also want to bring it to the referee's notice that all the scaling laws which we find to be close to linear are actually not exactly linear but are slightly slower than linear. As an example, for the case of one-hot coding as shown in Fig. S11, the power coefficient γ of the scaling law $N_c \propto K^\gamma$ is estimated as $\gamma = 1.08 \pm 0.01$.

In summary, as K increases, a smaller reconstruction error and larger than linear growing of N_c should be necessary to achieve a similar level of performance. The rather slow growth of the norm of w_{out} is truly intriguing, which suggests that the magnitude of the w_{out} entries do not need to grow large in order to distinguish the many attractors in the memory RC. This distinguishing task seems to be fulfilled by the modulation of the index value and a sufficiently large reservoir network.

Report of Reviewer #4

General comment: *“This paper discusses the development and application of reservoir-computing-based memories for storing and retrieving complex dynamical attractors. Traditional neural network models were limited to static patterns, while reservoir computing enables the handling of complex dynamic patterns, even chaotic attractors. The paper outlines two common recalling scenarios in neuropsychology: location-addressable with an index channel and context-addressable without such a channel. In the location-addressable retrieval scenario, a single reservoir computing machine can memorize numerous periodic and chaotic attractors, each retrievable with a specific index value. The paper provides control strategies for switching among these attractors and uncovers an algebraic scaling law between the number of stored attractors and the reservoir network’s size. In the context-addressable retrieval scenario, the multistability with cue signals is exploited. With the increase of the length of the cue signal, a high success rate can be achieved for attractor retrieval. The paper offers insights into developing long-term memories for complex dynamical patterns. The paper is well-structured, providing a clear introduction and explanation of the topic. However, the embedding multiple low-dimensional attractors in a single reservoir-computing system and achieving switching among them using external input cues has been proposed and explored to some extent in [1]. Thus, the following issues should be addressed.”*

Response: We appreciate that the referee considered our paper to “offer insights into developing long-term memories for complex dynamical patterns”, and that our paper is “well-structured” and “providing a clear introduction and explanation of the topic”. We also really appreciate the other comments of the referee and have made considerable extensions and revisions to our paper accordingly.

Comment 1: *“The article presents some cases where the switching between attractors failed. Did these failures lead to convergence to other embedded attractors or unknown attractors within the high-dimensional reservoir space, or was it due to the destination attractor not being effectively and stably learned? More explanations are required.”*

Response: Inspired by the comment of the referee, we have substantially added more results and discussions on the nature of failed switching trials. We find out that these failures, as the referee has speculated, lead to convergence to some (almost random) untrained attractors within the high-dimensional reservoir space. Following this result, we conducted an investigation into the pattern of transition matrices, the basin structure of index-based memory RC, and how these two aspects are connected. The newly added discussions are as follows, accompanied by the new panels of Figs. 2(H) and 2(I) as illustrations (on pages 5-6)

- Examining the transition matrices, we find an interesting pattern. As shown in Fig. 2(F), the variance of the success rate across different columns (i.e., different destination states) is much larger than the variance across different rows (i.e., different starting states). This implies that the success rate is significantly more dependent on the destination attractor than on the starting attractor.

Why, then, is there such an asymmetric dependence in the transition matrices? And more essentially, what is the dynamical mechanism behind switch failures? Before attacking these questions directly, we need to understand several points. We first notice the dynamic consequence of a switch in the index value \mathbf{p} . Incorporating this term of \mathbf{p} into the reservoir computer is equivalent to adding an adjustable bias term to each neuron in the reservoir hidden layer. Different values of \mathbf{p} thus directly result in different bias values

on the neurons and, thus, different dynamics in each neuron. The same indexed RC under different p can be treated as different dynamical systems with different dynamical equations. Also, notice that during such a switch, we do not directly interfere with the state input u or the hidden state r of the RC network, but only change the value of p . Therefore, a switch in p , like the ones we did in Fig. 2(A), is switching the dynamical equation of the RC network while keeping the last time step of the RC states. More specifically, the last time step of the hidden state r_{last} and state output v_{last} are passed from the previous dynamical system to the new dynamical system after switching. This pair of r_{last} and v_{last} becomes the initial (hidden) state and initial input under the new dynamic equation of RC.

Following these understandings, we can plot the basin of attraction of the trained state under its corresponding index value. We find that, unfortunately, the basin of attraction of the trained state usually does not fully occupy the entire phase space of the RC network. In Fig. 2(I), we show the basin structure of two arbitrary states in an indexed RC trained with 16 attractors. The blue regions, leading to the trained states, are leaving some space for the orange regions, which lead to untrained states. After some verification, we confirm that these basins to untrained states are the origins of failed switching. If the RC states of the last time step before switching, which become the initial states after the switching, lay outside the blue region, then the RC network will evolve to an unwanted state, and a switch failure will occur. We can further plot the points from the attractor before switching with blue dots (successful) and orange dots (unsuccessful), as shown in Fig. 2(H). They are the projections of the basin structure of the new attractors onto the previous attractor. Furthermore, this understanding can help explain the strong dependence of success rates on the destination states. Now, we can see that two factors determine the success rate: the relative size of the basin of the new state after switching and the degree of overlap between the previous attractor and the new basin. While the degree of overlap depends on both the starting and destination states, the relative size of the new state basin is solely determined by the destination state. Therefore, we should expect a stronger dependency of the switching success rate on the destination state. Further illustrations of (i) the basin structure of the memorized states in an indexed RC, (ii) projections of the basin structure of the destination state back to the starting state attractor, and (iii) how switch success is strongly affected by the basin structure of the destination state are demonstrated in the SI.

The other possibility that the referee mentioned, which is “due to the destination attractor not being effectively and stably learned”, should also be possible. However, we consider that scenario to be a case where the memory RC is not well-trained. Thus we have not focused on that case. With proper tuning on the hyperparameters and sufficiently large reservoir network, such unstable memory states are rare to find in our index-based memory RCs.

Comment 2: *“Intuitively, the level of difficulty in switching attractors is expected to increase with the increase of the number of the embedded attractors. Is there a correlation between the success rate of switching and the quantity of embedded attractors?”*

Response: This is an interesting problem and is more complicated than it might appear to be. Counter-intuitively, we find that increasing the number of embedded attractors might not always lead to worse success rates (assuming all other factors remain the same). A closely relevant (and also arguably counter-intuitive) phenomenon is that increasing the size of the reservoir network, which can usually help us have a better quantity of embedded attractors, can be harmful to attractor switching in certain cases. Behind these two phenomena is an intriguing and possibly generic relationship between the reservoir network size and the basin structure

in the hidden space of RC. Our preliminary results suggest that there exists a critical dimension of RC for a set of attractors to memorize, and this dimension is determined by many features of each attractor and among different attractors. Given the complexity of this problem, we believe a separate paper is required to provide a comprehensive understanding of this issue.

Comment 3: *“It is interesting that a scaling law between the number of attractors and the network size is uncovered. However, this article only considers the case of 3-dimensional attractors. I am curious to know if the parameter ν of the scaling law is related to the dimensionality or the box-counting dimension of the embedded attractors. Furthermore, I wonder whether or not there is an upper bound on the memory capacity of reservoir-computing to store attractors.”*

Response: Inspired by the advice of the referee, we have substantially added more results in the scaling behavior part, with different sets of target states and different RC structures or training schemes. These results can be found in Fig. 4 of our new version of the paper. We indeed find a variety of different scaling behaviors other than a linear law. However, this variety also unveils the previously hidden complexity of the phenomenon. We believe it should be very difficult to draw a universal relationship between the features of the attractors and the scaling law. As an example, we find that for index-free memory RCs, factors such as the degree of overlap among the target states can significantly affect scaling behavior.

Comment 4: *“The article utilizes validation errors over a period of time to assess whether or not the attractors have been successfully memorized. However, the objective of this study is long-term memory rather than short-term memory. Therefore, it might be more appropriate to select indicators that better characterize long-term patterns of the systems, such as the maximum Lyapunov exponent, to determine the success of memory retention.”*

Response: We agree that using these indicators to measure the long-term fidelity of attractor reconstruction can provide strong support to our approach to multifunctional RC. We have added results with the maximum Lyapunov exponents and correlation dimensions in the main text and the SI. These new results can justify that the long-term features of the memorized chaotic state (the “climate”) can be faithfully reproduced during memory retrieval. In the main text, we added the following discussion (in the left column of page 4):

- We evaluate the fidelity of the recalled states in the long term by calculating the maximum Lyapunov exponents and correlation dimension. We train an ensemble of 100 different memory RCs, recall each of the memory states, and generate outputs continuously for 200,000 steps. The two measures are calculated within this generation process and compared with the ground truth. As an example, for the Lorenz attractor, with a ground truth correlation dimension of 2.05 ± 0.01 , in 96% of the memory RCs, we retrieve an attractor with a correlation dimension within 2.05 ± 0.02 . As for the maximum Lyapunov exponent, the ground truth for the Lorenz attractor is about 0.906, and 91% of the memory RCs fall in the small range of 0.906 ± 0.015 . A more comprehensive result is shown in the SI, with similarly high fidelity for all other states. This remarkable result shows that it is indeed the long-term “climate” that the memory RC has learned and can reproduce.

In Supplementary Note 2, we further provide results of the maximum Lyapunov exponents of the reconstructed chaotic attractors, and demonstrate how this performance varies across different levels of training noise

σ_{noise} . The results are demonstrated in Fig. S3, showing that within an optimal noise interval, the maximum Lyapunov exponents of the reconstructed chaotic attractors agree well with the ground truth values.

However, we would like to point out that this issue of fidelity of long-term patterns is different from the issue of whether a memory device can have long-term memory, though they are both related to a relatively long time scale. It is also indeed true, though, that our framework satisfies both “long-term” criteria. So we understand that this can be confusing. The former criterion requires triggering memory recalls that persist for a long term without losing the crucial dynamical features of the target memory state. The latter criterion requires that the dynamical information is stored in the weights and connections of the Rc network, not the hidden state, so that the memory states can be recalled with proper cues or other recalling methods, even with a random initial hidden state. In other words, the “term” in the name “long-term memory” actually refers to the time *between* memorizing and recalling, not to the time length *of* recalling. To make this point more clear to readers, we have added the following discussion in Supplementary Note 2:

- Note that this “fidelity of recalled attractors in the long term” is a separate issue from achieving “longterm memory”, although both expressions have something to do with a long time scale. It is also indeed true, though, that our framework satisfies both “long-term” criteria. The former criterion requires that once a memory state is recalled, it can persist for a long term without losing or deviating from the crucial dynamical features of the target memory state (such as the maximum Lyapunov exponent). The latter criterion requires that the dynamical information is stored in the weights and connections of the Rc network, not the hidden state, so that the memory states can be recalled with proper cues or other recalling methods, even with a random initial hidden state. Thus, a recalled state that only persists in the memory device for several periods may still be considered as “long-term memory”. In other words, the “term” in the name “long-term memory” actually refers to the time *between* memorizing and recalling, not to the time length *of* recalling.

Comment 5: “The article introduces three effective control strategies to enhance the success rate of attractor switching. However, there is no accompanying data in the study to clearly demonstrate the specific improvement each method brings to the success rate.”

Response: In the revised version, we have added results with performance measurements of our three control strategies. We have also demonstrated how these performances would change with respect to different parameter values in the strategies, such as different maximum detour numbers in the detour-based approach or different levels of random perturbation in the feedback control approach. In the main text, these new results are mainly demonstrated in Figs. 3(B), 3(E), and 3(F) with associated discussions in the main text, including the following three paragraphs:

The Detour Method (in the right column of page 6):

- This method can be relatively simple to implement in many scenarios, as all we need to do is just to switch the p value a few more times. However, this relatively simple strategy has two limitations. First, it is necessary to know some information about the transition matrix to search for an appropriate detour and estimate the success rate. Second, our understanding of the dynamical mechanism behind failed switching suggests that the success rate mostly depends on the relative size of the attracting basin of the destination state. This implies that for a state that has a small attracting basin, it should be difficult to reach from almost all other states. Thus, the improvement of the success rate from a detour can sometimes be

limited to below reaching 100%. As exemplified in Fig. 3(B), the success rate with detours, which can significantly increase the overall success rate within just a few steps, can saturate at levels lower than 1.

The Feedback Control Method (on pages 6-7):

- To provide a quantitative estimation of the performance of this strategy, we run 90,000 switching in 25 different memory RCs. These memory RCs are trained with the six periodic and chaotic states, as illustrated in Fig. 1(B). The size of the reservoir network is 1,000, and other hyperparameters are listed in the SI. Among these 90,000 switchings, 8,110 trials (about 9%) failed, which we used as a pool to test the control strategies. To make our results relatively generic, we test 12 different control settings with different parameters. We test four different lengths of each noisy perturbation period, including 1 step, 3 steps, 10 steps, and 30 steps. We test Gaussian white noise with three different noise levels, with standard deviations of $\sigma = 0.3$, $\sigma = 1$, and $\sigma = 3$. We find that with an appropriate choice of the control parameters, 10 periods of 10 steps of noise perturbations (so 100 steps of perturbations in total) can eliminate 99% of the failed switching, as illustrated in Fig. 3(E). The full results of the 12 different control settings are demonstrated and discussed in the SI.

The Cue-Assisted Method (in the left column of page 8):

- With this third control strategy with the cues, we again employ the 8,110 failed switching pool to test the strategy performance. We find that, as shown in Fig. 3(F), with the increase of the cue length, the success rate increases nearly exponentially, approaching a nearly random-access scenario. Almost 90% of the failed switchings can be eliminated with just 8 steps of cue, and 24 steps of cue can eliminate more than 99% of the failed switchings.

Moreover, in the SI, we show how the performance of the feedback control method changes with respect to the choice of different parameters, including the magnitude and length of the random perturbations and the maximum number of trial-and-error iterations. The performance of the method under these various conditions is shown in Fig. S9.

Comment 6: *“For the feedback control, an asymmetric distance between each pair of stored attractors is proposed, which is associated with the time duration that the random perturbation lasts. However, the article does not explicitly specify the length of the random signals required for switching among attractors, which is a crucial metric for evaluating the effectiveness of this control strategy.”*

Response: We apologize for this overlook and have added this information to the paper. Furthermore, we extend our results to various scenarios with different time duration of the random perturbations, as shown in Fig. 3(E) and Fig. S9. Our new results suggest that as short as 3 steps of random perturbation can already result in a high success rate within a few time of perturbation. We also find that the performance enhancement saturates at around 10 steps of perturbation length, and longer perturbation will not significantly affect the results.

Comment 7: *“Is this type of reservoir network able to apply the time-series prediction, such as [2] ?”*

Response: This is an interesting point. Our memory RC can indeed make time-series predictions. With a short cue, our memory RCs can be activated with long and persistent dynamical pattern, which should be considered as extending or continually predicting the short input cue. However, we do not think that our memory RCs, without significant modification to the structure or training scheme, can reach the level of prediction results as

the cited paper. It is worth combining the ideas of index channels or the coexistence of multiple states with the ARNN discussed in that paper. We have cited this paper.

REVIEWERS' COMMENTS

Reviewer #1 (Remarks to the Author):

The authors have very thoroughly addressed all issues I raised in my review and worked out extensive and convincing answers. Consequently, they added significant new material to the paper (and to the SI). I am fine with the new version of the manuscript and can now recommend the paper for publication.

Reviewer #2 (Remarks to the Author):

The authors have addressed the questions and remarks given in the referee reports. They have clarified technically aspects of their work which were necessary to fully understand their results. Importantly, they have expanded their results section on the scaling laws and demonstrated a scenario in which the approach of training one large network to memorize many attractors appears more efficient than training many separate small reservoirs. I recommend the paper to be published.

Reviewer #3 (Remarks to the Author):

I commend the authors on a spectacular, thorough, and rigorous revision. Not only have they satisfactorily answered all of my comments, but I believe their extended results and discussion have significantly raised the impact and interest of their manuscript within and outside of the reservoir computing community. I recommend acceptance.

Point-by-point response to referee comments

We are grateful that the referees recommended our revised manuscript for publication. There are no specific comments to address from this round of review.

Report of Referee #1

Comment: *“The authors have very thoroughly addressed all issues I raised in my review and worked out extensive and convincing answers. Consequently, they added significant new material to the paper (and to the SI). I am fine with the new version of the manuscript and can now recommend the paper for publication.*

Response: Thanks a lot and we are grateful for the insightful comments provided earlier.

Report of Referee #2

Comment: *“The authors have addressed the questions and remarks given in the referee reports. They have clarified technically aspects of their work which were necessary to fully understand their results. Importantly, they have expanded their results section on the scaling laws and demonstrated a scenario in which the approach of training one large network to memorize many attractors appears more efficient than training many separate small reservoirs. I recommend the paper to be published.*

Response: Thanks a lot and we are grateful for the insightful comments provided earlier, especially on the issue of training one large network versus many separate small reservoirs.

Report of Referee #3

Comment: *“I commend the authors on a spectacular, thorough, and rigorous revision. Not only have they satisfactorily answered all of my comments, but I believe their extended results and discussion have significantly raised the impact and interest of their manuscript within and outside of the reservoir computing community. I recommend acceptance.*

Response: Thanks a lot and we are grateful for the insightful comments provided earlier.